# m6a methylation orchestrates IMP1 regulation of microtubules during human neuronal differentiation

Pierre Klein [1,2], Marija Petrić Howe[2,3], Jasmine Harley[2,3], Harry Crook[2], Sofia Esteban Serna [1], Theodoros I. Roumeliotis [4], Jyoti S. Choudhary [4], Anob M. Chakrabarti [5], Raphaëlle Luisier[6,7], Rickie Patani [2,3] ✉ & Andres Ramos [1] ✉

Neuronal differentiation requires building a complex intracellular architecture, and therefore the coordinated regulation of defined sets of genes. RNA-binding proteins (RBPs) play a key role in this regulation. However, while their action on individual mRNAs has been explored in depth, the mechanisms used to coordinate gene expression programs shaping neuronal morphology are poorly understood. To address this, we studied how the paradigmatic RBP IMP1 (IGF2BP1), an essential developmental factor, selects and regulates its RNA targets during the human neuronal differentiation. We perform a combination of system-wide and molecular analyses, revealing that IMP1 developmentally transitions to and directly regulates the expression of mRNAs encoding essential regulators of the microtubule network, a key component of neuronal morphology. Furthermore, we show that m6A methylation drives the selection of specific IMP1 mRNA targets and their protein expression during the developmental transition from neural precursors to neurons, providing a molecular principle for the onset of target selectivity.

During development, neurons establish inter-cellular networks that acquire, retain and respond to information in a spatiotemporally regulated manner. The architectural development of neurites and synapses, and the underlying cytoskeletal changes, require the regulation of gene expression programs by neuronal RNA-binding proteins (RBPs). While the action of these proteins on individual targets has been studied in detail, our understanding of how RBPs regulate networks of genes and orchestrate cellular processes during neuronal differentiation remains limited.

IMP1 is a well-studied RBP that regulates the localisation, stability and translation of mRNAs[1] and is essential for embryonic development[2].

In developing neurons, IMP1 plays a key role in establishing neurite outgrowth and synaptogenesis[3–5]. Molecular studies have focused on its physical and functional interactions with a small number of mRNA targets. In particular, the interaction between IMP1 and beta-actin mRNA has been used as a model system to explore the concept of an RNA 'zipcode' in the regulation of mRNA transport and translation in the cell[6–8]. IMP1 transcriptome-wide binding and motif analyses have previously been performed, but only in highly proliferative cells including cancer and human pluripotent stem cells[9,10]. These studies have highlighted a role for IMP1 in cellular adhesion, proliferation and survival. However, the mechanism by which IMP1 regulates global gene networks

[1]Division of Biosciences, Institute of Structural and Molecular Biology, University College London, Darwin Building, Gower Street, London WC1E 6XA, UK. [2]Human Stem Cells and Neurodegeneration Laboratory, The Francis Crick Institute, 1 Midland Road, London NW1 1AT, UK. [3]Department of Neuromuscular Diseases, Queen Square Institute of Neurology, University College London, London WC1N 3BG, UK. [4]Functional Proteomics team, The Institute of Cancer Research, 237 Fulham Road, London SW3 6JB, UK. [5]RNA Networks Laboratory, The Francis Crick Institute, 1 Midland Road, London NW1 1AT, UK. [6]Idiap Research Institute, Martigny 1920, Switzerland. [7]SIB Swiss Institute of Bioinformatics, Lausanne 1015, Switzerland. ✉e-mail: rickie.patani@ucl.ac.uk; a.ramos@ucl.ac.uk

underlying the establishment of neuronal architecture remains unclear and is a critical knowledge void in neuronal development.

Here, we examine the system-wide role of IMP1 during the differentiation of human neurons, to elucidate how this paradigmatic protein is developmentally regulated and gain insight into how it controls the expression of essential gene programs. We show that IMP1 transitions to a different set of targets during neuronal specialization. This developmental transition is regulated by an increase of m6A methylation and, in turn, underlies the regulation of protein expression. Our findings establish IMP1 as an important regulator of the microtubule network during neuronal development.

## Results

### IMP1 transitions to a new set of neuronal targets in a regulated fashion during differentiation

RBPs have cell-type specific roles determined by the mRNAs to which they bind and the specific sites of protein-RNA interaction. IMP1 plays

multiple roles that are essential for the development of the nervous system, including synaptogenesis, dendritic arborisation and axonal pathfinding, among others[4,11]. While IMP1 interaction with *ACTB* mRNA underlies an important function in neuronal development, its global interactome and role in gene regulation in neurons have not yet been explored. An essential question is whether the regulation of complex morphological neurodevelopmental processes requires IMP1 to bind to a broader set of RNAs than previously recognized. Notably, transcriptome-wide data are largely limited to highly proliferating cells, where IMP1 predominantly interacts with non-neuronal pathways.

Here we have used a human induced pluripotent stem cell (hiPSC) differentiation model, and interrogated the developmental transition from neural precursor cells (NPCs) to (isogenic) neurons. In this process, differentiating NPCs undergo major morphological changes to become neurons by building a network of neurites and forming synapses (Fig. 1a, b and Supplementary Fig. 1a, b). Firstly, we confirmed the predominantly cytoplasmic expression of IMP1 in these two stages

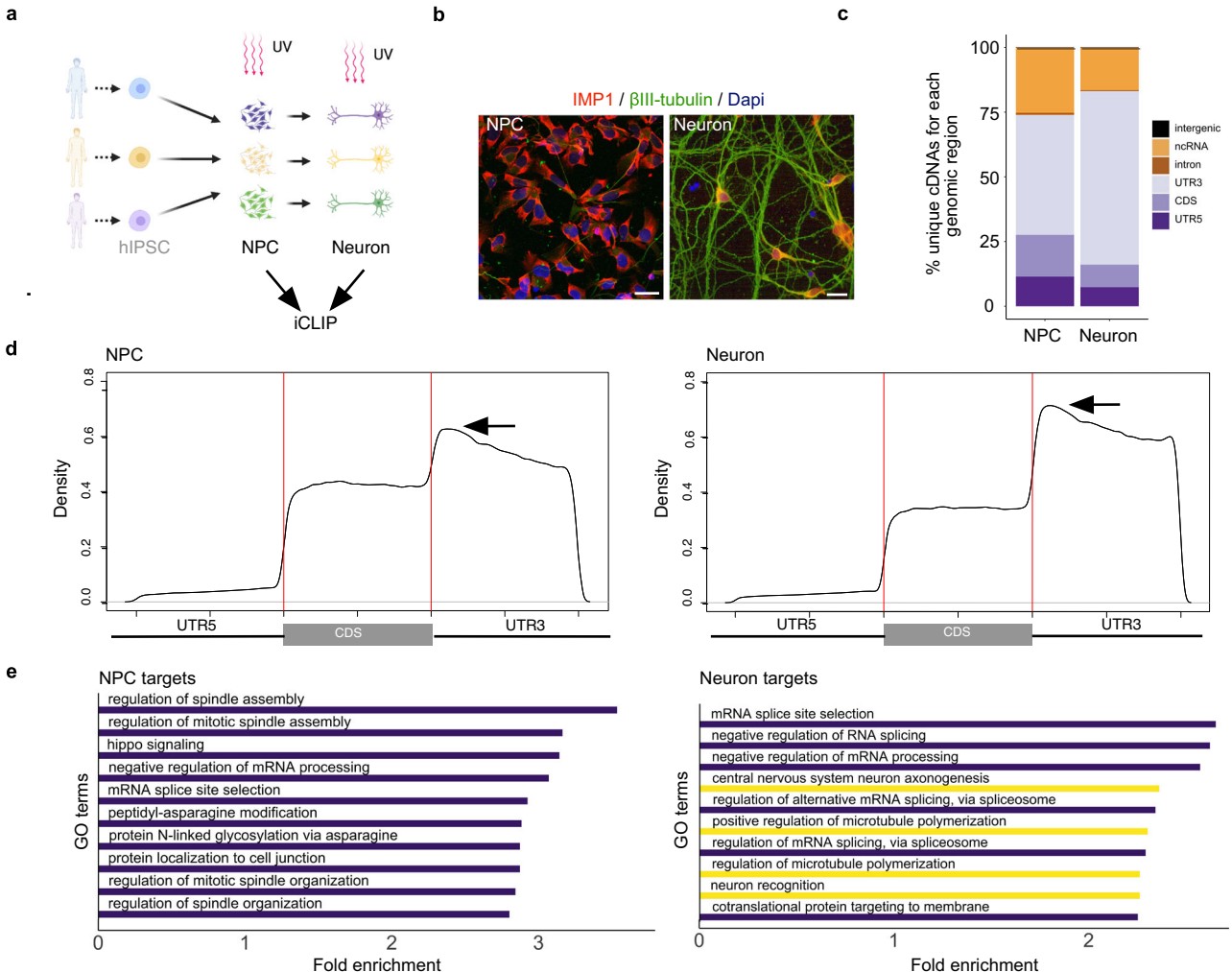

**Fig. 1 | IMP1 transitions to neuronal targets during differentiation from NPCs. a** In our iCLIP assays, NPCs and neurons were differentiated as previously described[27]. Three independent iPSC lines (biological replicates) were used for both NPCs and neurons. Image created with BioRender.com released under a Creative Commons Attribution-NonCommercial-NoDerivs 4.0 International license. **b** Representative confocal images of NPCs and neurons immunolabeled for IMP1 and βIII-tubulin, and counterstained with DAPI. Scale bar is 20 μm. *n* = 3 independent iPSC lines. **c** Percentage of unique cDNAs mapping to each region of the transcriptome in neurons and NPCs. Counts were normalized to the number of genomic nucleotides corresponding to each region, and percentage of total counts was calculated. Percentage is as follows: NPCs−5′UTR 11.44, CDS 16.13, 3′UTR 46.36,

Intron 0.81, ncRNA 25.13, Intergenic 0.13; MN − 5′UTR 7.32, CDS 8.71, 3′UTR 67.12, Intron 0.37, ncRNA 16.37, Intergenic 0.11. **d** Metagene plots of IMP1 crosslink density in NPCs (left) and neurons (right). Vertical lines mark the start and the end of the CDS. Arrows highlight that, while the protein transitions to the 3′UTR in neurons, the distribution within the 3′UTR appear to be similar in neurons and NPCs. **e** Top ten GO terms in neurons and NPCs, sorted by fold enrichment. Analysis was performed using PANTHER. All terms displayed have a false discovery rate (FDR) < 0.05 and *p* value < 0.01. *P* value was calculated using a two-sided Fisher's Exact test. Terms linked to neurogenesis or microtubules are highlighted in yellow. Source data are provided as a Source Data file.

of development and its presence in developing neurites and in synapses, the latter labelled using pre- and post-synaptic markers SYT-1 and Homer-1 (Fig. 1b, and Supplementary Fig. 1b–d). Next, we sought to define the dynamic regulation of the IMP1 protein-RNA interactome during neuronal differentiation. To this end, we scaled up our model of neuronal differentiation to millions of cells, and performed IMP1 individual-nucleotide resolution UV crosslinking and immunoprecipitation (iCLIP) on both NPCs and neurons (Fig. 1a and Supplementary Fig. 2a–h) yielding an average of 3 million unique reads in NPCs and 4 million in neurons.

IMP1 crosslink sites were predominantly located in 3′UTRs, as previously reported in CLIP studies of highly-proliferative and cancer cells[9,10]. Importantly, comparison of binding at the two stages of lineage restriction revealed that the IMP1-binding landscape is developmentally regulated, increasing its binding to the 3′UTR from 45 to 70% during the transition from NPCs to neurons (Fig. 1c, d). We also examined whether neuronal IMP1 binding sites were enriched in the consensus motifs of IMP1 individual KH domains using a HOMER-based approach. Our analyses used a sequence of 5 bases, given that KH domains can recognise specifically up to 5 nucleotides. An initial analysis performed with a narrow 10-nucleotide window identified a highly enriched KH4 core consensus sequence (GGA)[12,13], while a follow-up analysis with a broader 30-nucleotide-window, instead, returned three motifs. The two highest scoring among these (CCGTT and ACACA) contain the core consensus sequences of KH1 (CC(or G) G)[14] and KH3 (CA/ACA)[12,13] (Supplementary Fig. 3a). In NPCs, although the results were less clear-cut, the KH4, KH3 and KH1 motifs were also present (data not shown). Notably, a CA sequence has been previously reported to be enriched in IMP1 target sites in previous computational studies[10]. These results connect IMP1 neuronal target recognition with the current molecular understanding of the protein's target specificity. Next, a comparison of mRNAs interacting with IMP1 in NPCs and neurons indicated that the protein binds to a large subset of targets, a substantial proportion of which are developmental stage-specific (Fig. 1e and Supplementary Fig. 2e, g). Indeed, a Gene Ontology (GO) analysis at the two developmental stages identified pathways related to early neuronal differentiation processes such as spinal cord patterning, chromatin re-organisation and synaptic development in the NPCs, while in neurons revealed pathways related to later stages of neuronal development such as axonogenesis, synapse maturation and microtubule polymerisation (Fig. 1e and Supplementary Fig. 2h).

Neuronal differentiation is accompanied by a transcriptional upregulation of a large set of genes. An important mechanistic question is whether an additional process of IMP1 target specialisation is taking place that may be important in the regulation of neuronal genes. To address this, we compared IMP1-RNA binding profiles on individual transcripts, where changes in gene expression during differentiation can also be directly compared by re-analysing our prior RNAseq data (Fig. 2a and Supplementary Fig. 2f)[15].This comparison highlighted that, in many cases, an increase in binding is not necessarily linked to an increase in gene expression. Indeed, when considering the transcripts that our iCLIP experiments identify as neuronal IMP1 targets (2657), 1218 are upregulated and 722 are downregulated in neurons when compared to NPCs, as derived from data shown in Supplementary Fig. 2g. Next, we explored differential binding at the transcriptome-wide level in NPCs and neurons, normalizing the changes in IMP1 peaks for RNA transcript abundance. This showed that while transcript abundance is one important determinant of IMP1-mRNA binding in neurons, the increase in a large (2335) group of iCLIP peaks is independent of relative mRNA abundance (Fig. 2b). Notably, a much smaller group of peaks (868) exhibit decreased binding. This asymmetric distribution is further accentuated when considering the peaks on mRNAs encoding proteins related to microtubule organisation and axonal development (Supplementary Fig. 3b), confirming that the increase in IMP1 binding to these mRNAs cannot be justified by their increased expression alone. This is consistent with the results of the GO analysis of the developmentally regulated peaks (Fig. 1e, Supplementary Fig. 3c), which revealed an enrichment of microtubule-related mRNAs that are regulated by IMP1. Importantly, the increase in IMP1 occupancy of these targets cannot be explained by an increase in the abundance of IMP1, as our data indicate that the abundance of IMP1 in the cell decreases in the transition from progenitors to neurons, in accordance with its expression decreasing during the development of brain and other tissues[2,16] (Fig. 2c, Supplementary Fig. 3d). Finally, we probed IMP1 function by examining neurite branching and synaptic development, which have previously been associated with IMP1[3,4]. To directly address this, we performed a morphological analysis of neurite branching, revealing a significant decrease in complexity in IMP1 knockdown (KD) neurons compared with their control counterparts (Fig. 2d). Furthermore, we showed that IMP1 KD decreases the size (but not the number) of SYT-1 positive synaptic puncta (Supplementary Fig. 3e). Together, these data show that IMP1 regulates essential neurodevelopmental processes both at the molecular and cellular levels in differentiating human neurons.

## IMP1 regulates a network of microtubule genes during human neurodevelopment

IMP1 is reported to regulate the stability, and therefore the concentration, of a set of functionally related mRNAs in highly proliferating cells[7,9,10,17]. However, this function is not generalisable to the global IMP1 interactome[10]. In addition, while IMP1 has been reported to regulate the translation of ACTB mRNA by sequestering the mRNA in a folded conformation, data in highly proliferating cells indicate an association of the protein with translationally active ribosomes[9]. Furthermore, IMP1 up-regulates translation of mitochondrial mRNAs[18]. It follows that, in principle, IMP1 could regulate the expression of sets of neuronal targets both at the protein and the RNA level. In order to determine whether IMP1 regulates the expression of sets of genes during neuronal development and whether regulation occurs at the protein or RNA level, we used mass spectrometry to profile the proteome of NPCs and neurons in control and IMP1 knockdown conditions, and intersected these data with paired RNAseq.

In NPCs, a similar number of proteins are up and downregulated upon IMP1 KD, while in neurons twice as many proteins are downregulated compared to upregulated (Fig. 3a, b and Supplementary Fig. 4a), upon a similar KD (Supplementary Fig. 4b). IMP1 neuronal regulation was validated, in a representative subset of targets using quantitative immunocytochemistry and Western blot (WB) as orthogonal experimental approaches (Supplementary Fig. 4c–e). A global functional analysis of the regulated targets indicated that IMP1 regulation of neuronal specialization is organised in gene networks underlying important cellular processes with two of the most highly represented categories of IMP1-regulated targets being related to synapses and microtubule regulation (Fig. 3c). These also include a small number of proteins involved in myristoylation. The microtubule related-proteins include tubulins, and regulators of microtubule stability such as kinases and essential microtubule-binding proteins. This suggests that IMP1 plays a coordinated role at multiple levels of microtubule regulation (Fig. 3c). An unbiased analysis of the relationship between IMP1-regulated proteins revealed that they interact both physically and functionally, creating a connected network (Supplementary Fig. 4f). This global and quantitative analysis of IMP1 regulation in neurons indicates that, in addition to moving along the microtubule system to organise local mRNA translation, as previously reported[1,8,19], IMP1 regulates the microtubule network itself.

Notably, analysis of RNAseq data in both control and IMP1 KD conditions showed that only a modest number of genes are significantly up or down-regulated, and that IMP1-mediated regulation of the microtubule network occurs at the protein (rather than mRNA) level (Fig. 3d). This is evidenced by discordant changes in IMP1 protein

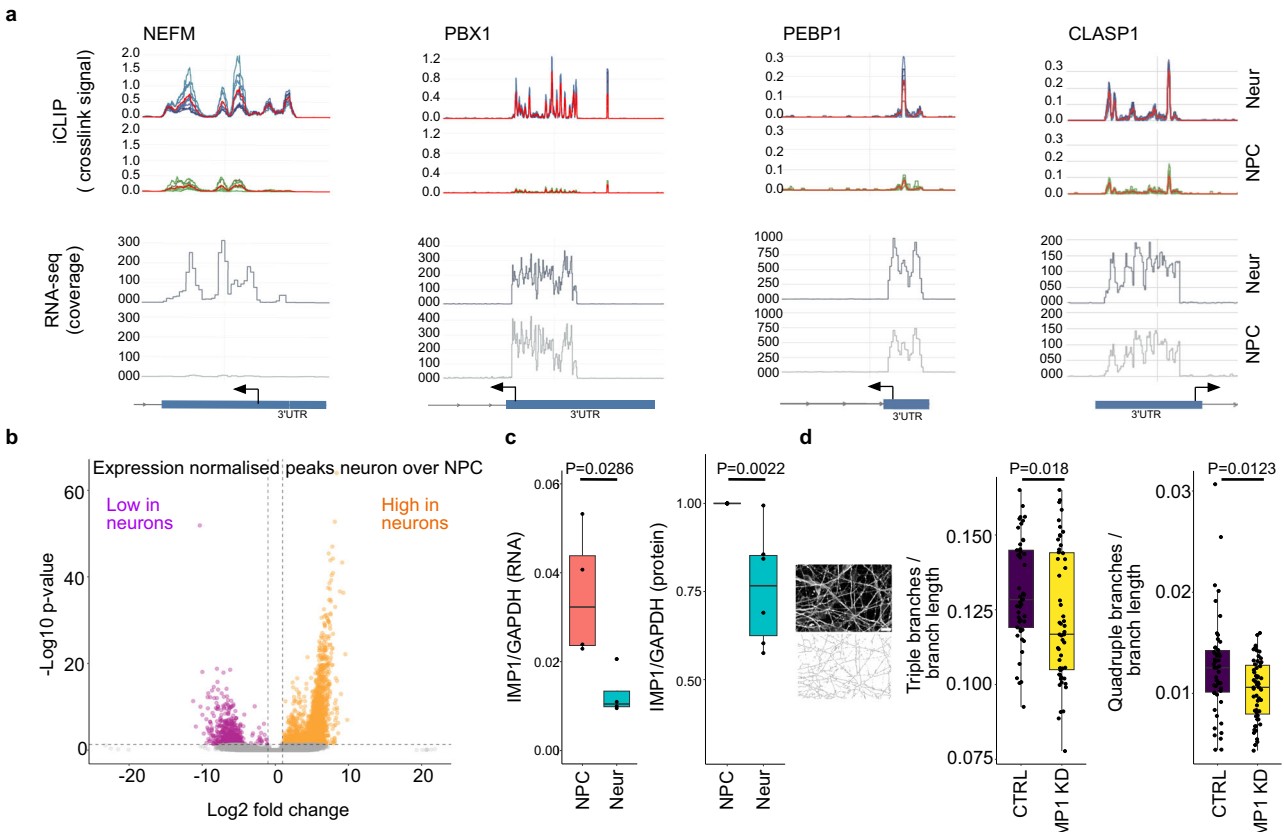

**Fig. 2 | IMP1 binding to neuronal mRNAs correlates with both a rise in the mRNA level and, independently, an increased recruitment of IMP1 to a subset of sites. a** IMP1 iCLIP crosslinks (counts per millions) and RNAseq data (sequencing reads) are mapped onto genomic coordinates in NPCs and neurons for 4 exemplar target mRNAs. iCLIP replicates are shown in blue for neurons and in green for NPCs, while the merged signal is in red. 3' untranslated region (3'UTR) and last exon are marked by a blue box and separated by an arrow, while line arrows indicate other intronic boundaries. In many of the RNA targets, the increase of IMP1 recruitment is not correlated with mRNA level. **b** Volcano plot of IMP1-bound peaks normalized by gene expression changes in NPCs vs neurons. Peaks with significantly higher signal in NPCs vs neurons, and vice versa are shown respectively in pink and orange. *P* values were calculated using a two-sided likelihood-ratio test (LRT). A threshold of 1 <Log2 Fold Change < −1 and adjusted *p* value < 0.05 was used to determine significance. For iCLIP, *n* = 3 independent iPSC lines for each condition (biological replicates); for NPCs, *n* = 2 technical replicates from 2 independent iPSC lines + 3 technical replicates from 1 independent iPSC line (biological replicates); for RNA-seq *n* = 3 independent iPSC lines for each condition (biological replicates). The vertical dashed lines denote log2 FCs < −1 or >1, while the horizontal dashed line marks an adjusted *p* value < 0.05. **c** Relative expression of IMP1 over GAPDH at NPC

and neuronal stages by RT-qPCR (RNA level, left) and WB (protein abundance, right). Data presented as boxplots, where the centre line is the median, limits are the interquartile range and whiskers correspond to 1.5 times the interquartile range. Outliers are not displayed for clarity. For the qPCR experiment *n* = 4 data points were obtained from independent iPSC lines (biological replicates), while for the WB experiment *n* = 6 data points (6 replicates, including 4 biological replicates (independent iPSC lines) in 2 experimental blocks). Values are normalized by the relative expression in the NPCs of the corresponding clones. *P* values were calculated using a two-sided Mann−Whitney U test and reported on the plot. **d** Left−representative image of βIII-tubulin (top) and skeletonization (bottom) used for the branching analysis. Approximated scale bar is 20 μm. Middle and right−number of triple and quadruple branches in neuronal processes normalized against branch length, in IMP1 siRNA (IMP1 KD) vs non-targeting siRNA control (CTRL). Data presented as boxplots where the centre line is the median, limits are the interquartile range and whiskers correspond to 1.5 times the interquartile range. Outliers are not displayed, for clarity. Data points represent different fields of view, *n* = 3 independent iPSC lines (biological replicates) in 3 independent experiments. *P* values were calculated using two-sided Mann−Whitney U test and reported on the plot. Source data are provided as a Source Data file.

and RNA targets upon KD (Fig. 3d). Indeed, at the global level more proteins are downregulated than upregulated in neurons while this is not the case for the corresponding mRNAs. The mode of regulation we observe is therefore different from the RNA stabilisation function of IMP1 reported in cancer cells and other cell lines[7,9,10,17].

In order to establish whether the effect of IMP1 on the proteome is direct, we re-examined the IMP1-regulated proteome in NPCs and neurons considering only RNA targets that were directly bound by the protein using our iCLIP data. We found a striking enrichment (to a 5-fold ratio) in the proportion of downregulated compared to upregulated proteins encoded by IMP1 RNA targets upon IMP1 KD in neurons (Fig. 4a, b), indicating that the regulation of protein expression discussed above is a direct one. Interestingly, in NPCs we also observed a modest 2-fold increase in the downregulated compared with upregulated targets upon IMP1 KD (Fig. 3b and Supplementary Fig. 4g).

Beyond linking protein regulation to IMP1 binding, the changes in downregulated targets also indicate that a functional specialisation of IMP1 takes place during neuronal development.

A more in-depth analysis of the IMP1 binding pattern showed that targets positively regulated by the protein have a significantly higher number of binding sites compared to unchanged or negatively regulated targets (Supplementary Fig. 4h). Notably, in microtubule and cytoskeleton targets, this regulation is directly linked to the number of IMP1 peaks in a transcript (Fig. 4c). Consistently, the enrichment in IMP1 binding within the regulated targets is specific to microtubules and neuronal processes (Fig. 4d). Together, this highlights that the regulation of microtubule assembly is a key function of IMP1 in developing neurons and confirms that this function is mediated by the direct regulation of a large gene network, as illustrated by the relationship between IMP1-regulated microtubule factors (Fig. 4e).

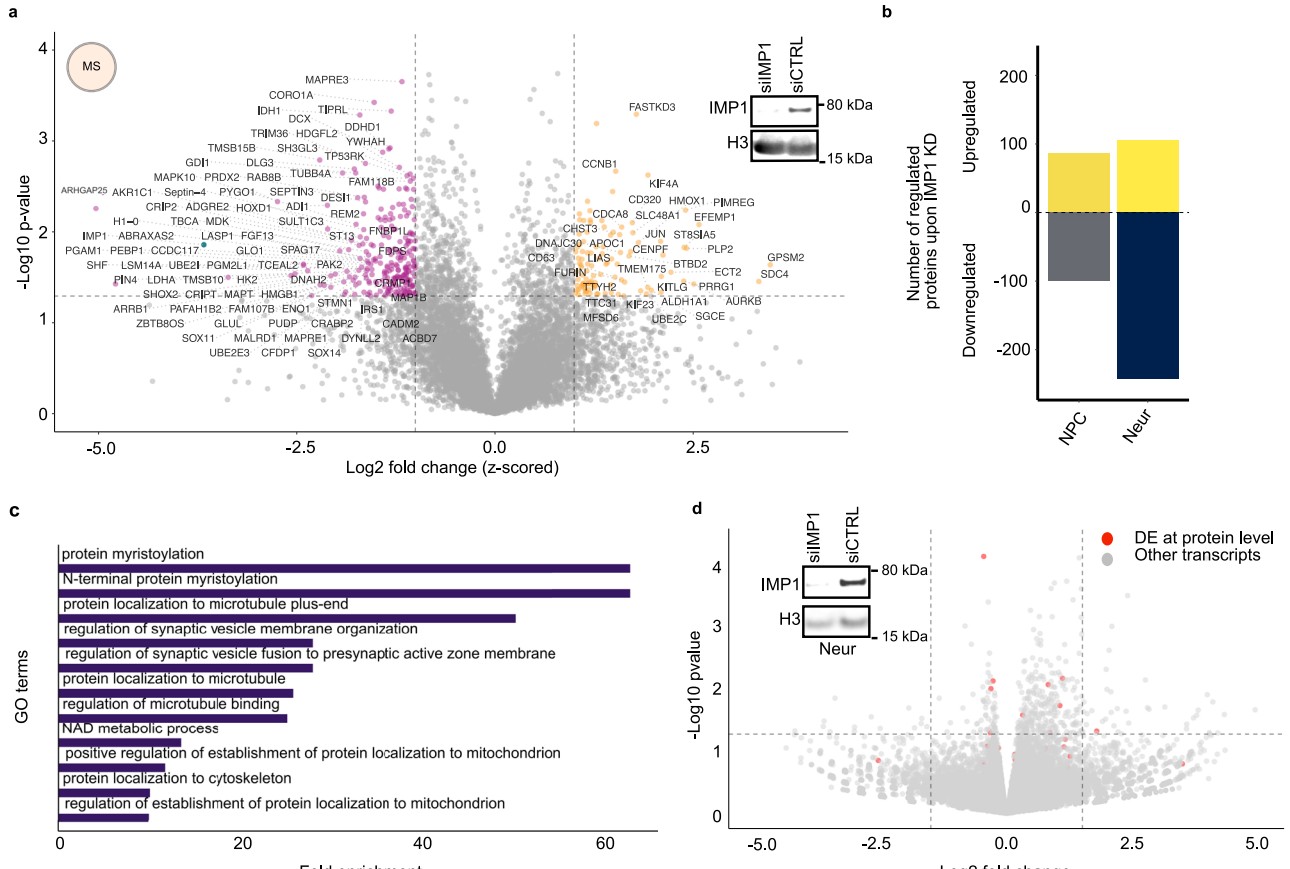

**Fig. 3 | IMP1 knock-down reduces the protein level of hundreds of neuronal genes. a** Volcano plot of proteins differentially expressed in IMP1 siRNA (silMP1) vs non-targeting control (siCTRL) neurons. Vertical dashed lines indicate log2 FC significance thresholds of 1 and −1, z-scored, while the horizontal dashed line denotes a p value significance threshold of 0.05. The proteins significantly up- or down-regulated are highlighted in orange and pink respectively. A two-sided one-sample Student's *t* test was used on *n* = 3 independent iPSC lines for each condition (biological replicates). The inset depicts a representative Western blot demonstrating IMP1 knock down (*n* = 3 independent iPSC lines for each condition), with H3 as loading control. **b** Numbers of up- and down-regulated proteins shown in (**a**) and Supplementary Fig. 4a, reported as a bar plot. *n* = 2 independent iPSC lines for each condition for NPCs. NPC neural precursor cells; Neur = neurons. **c** Top 11 GO terms in neurons ranked by fold enrichment. GO analysis was

performed using PANTHER. Terms displayed have a false discovery rate (FDR) < 0.05 and *p* value < 0.01 calculated using a two-sided Fisher's exact test. Only three proteins are found in the "protein myristoylation" and "N-terminal protein myristoylation" terms; two of these are protein phosphatases. **d** Volcano plot of mRNAs differentially expressed in neurons upon IMP1 knock down, from a comparison of RNAseq experiments in IMP1 knockdown vs control cells. Transcripts also shown to be regulated at the protein level by IMP1 knock down (see panel (**a**)) are depicted in red. Vertical dashed lines indicate log2 FC cut-off at 1.5 or − 1.5, while the horizontal dashed line indicates *p* value of 0.05. *P* value was calculated using a two-sided Wald test. *n* = 3 independent iPSC lines (biological replicates) were used for each condition. The inset depicts a representative Western blot demonstrating IMP1 protein silencing (*n* = 3 independent iPSC lines for each condition), with H3 as loading control. Source data are provided as a Source Data file.

## RNA methylation modulates IMP1 selection and regulation of the microtubule targets

Our data indicate that iCLIP peaks on a gene, and therefore IMP1 binding, change independently from RNA expression levels in a large set of targets (Fig. 2a, b), which implies an additional regulatory layer in IMP1 RNA target selection. Notably, the neuronal transcriptome has been reported to be highly enriched in m6A methylation[20–23]. In addition, IMP1 binding to m6A has been recently reported to regulate c-Myc and cell cycle targets in cancer cells[9]. To determine whether IMP1 regulation of the microtubule network during neuronal development is mediated by m6A methylation, we characterised the relationship between m6A methylation and IMP1 RNA binding during the NPC-to-neuron transition. While a number of recent studies have mapped m6A methylation to the transcriptome of human and mouse neurons, an m6A methylation atlas of the human transcriptome during neuronal differentiation has remained elusive. To directly compare the IMP1 RNA-binding peaks obtained from our iCLIP assays to neuronal m6A sites, we performed miCLIP on the aforementioned human NPCs and isogenic neurons (Fig. 5a, and Supplementary Fig. 5a). The miCLIP

dataset included an average of 1 million unique reads for NPCs and 4 million for neurons with high reproducibility of sites between replicates in both cell types (Supplementary Fig. 5b,c). As expected, the crosslink sites were enriched in the evolutionarily conserved DRACH consensus motif (D = A/G/U, R = A/G, H = U/A/C) and preferentially localised around the stop codon[20,22] (Fig. 5b and Supplementary Fig. 5d). It is noteworthy that comparing our data to 6 RNA methylation datasets from neural tissue spanning 3 species also showed a substantial overlap in m6A sites (21%) (Supplementary Fig. 5e).

Interestingly, starting from a similar quantity of RNA, we detected an increase in the number of m6A sites in neurons compared to NPCs (Fig. 5c). Consistent with this finding, we observed a concurrent increase in the level of the m6A methyl-transferase METTL3 together with the decrease in the level of the ALKBH5 demethylase (Supplementary Fig. 5f).

Importantly, we found the presence of both common and specific m6A sites in NPCs and their isogenic neurons, indicating a regulated role of methylation during neuronal differentiation (Fig. 5d). We then examined the IMP1-mediated regulation discussed in the previous

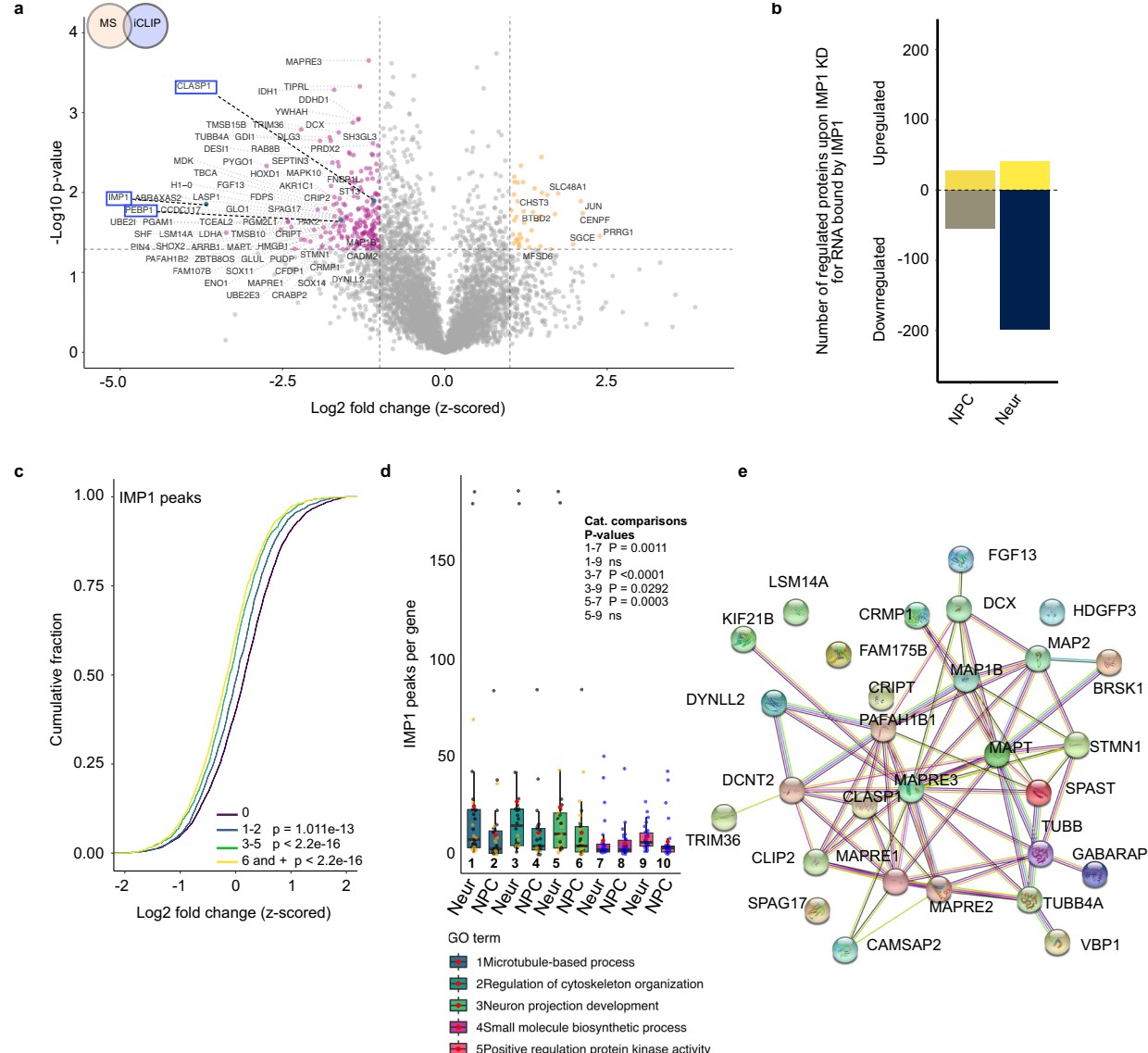

**Fig. 4 | IMP1 directly regulates a network of microtubule genes. a** Volcano plot showing proteins that are both differentially expressed upon IMP1 knock down in neurons and whose cognate mRNA is bound by IMP1 in our CLIP experiments. Proteins significantly up- or down-regulated in IMP1 siRNA (siIMP1) vs non-targeting control (siCTRL) are highlighted in orange and pink respectively. IMP1, CLASP1 and PEBP1 (Fig. 2a) are highlighted with blue dots and blue squares. Selection was based on log2 FC < −1 (down-regulated proteins) or log2 FC > 1 (upregulated proteins) and *p* value < 0.05. A two-sided one-sample Student's *t* test was performed, for transcripts containing at least one IMP1 binding site. For MS, *n* = 3 independent iPSC lines for each condition (biological replicates) were used; for iCLIP, *n* = 3 technical replicates from 3 independent iPSC lines (biological replicates) were used. **b** Number of upregulated and downregulated proteins in neurons, from (**a**) and NPCs, from Supplementary Fig. 4g, reported as a barplot. For MS on neuronal samples, replicates are as stated above. For MS on NPCs, *n* = 2 independent iPSC lines (biological replicates) for each condition were used; for iCLIP on NPCs, *n* = 2 technical replicates from 2 independent iPSC lines + 3 technical replicates from 1 independent iPSC line (biological replicates) were used. **c** Cumulative distribution plot of log2 FC in protein expression between IMP1 KD and control in neurons, as detected by MS, was plotted for RNA classes with 0, 1–2, 3–5 and 6 or more peaks in iCLIP experiments. For MS, *n* = 3 independent iPSC lines for each condition (biological replicates); for iCLIP, *n* = 3 technical replicates from 3 independent iPSC lines (biological replicates). The reported *p* values were calculated using a two-

sided Kolmogorov–Smirnov test comparing 1–2 peaks, 3–5 peaks or 6 and + peaks to 0 peaks. **d** Number of IMP1 peaks per gene detected by iCLIP for GO categories related to microtubules, neuronal or other pathways classified by PANTHER. Number of genes in each category is in the same range (between 26 and 30). Data presented as boxplots where the centre line is the median, limits are the interquartile range and whiskers correspond to 1.5 times the interquartile range. Outliers are not displayed for clarity. Red dots represent the mean. The categories "Cytoskeleton organization", "Microtubule-based process", "Neuron projection development" contain many common genes. Black dots represent transcripts which are present in more than one of these categories, while orange dots represent transcripts in a single category. "Positive regulation protein kinase activity" with "Small molecule biosynthetic process" include different transcripts, represented by blue dots. Each category is represented by a number below the corresponding bar. *P* values were calculated using a two-sided Mann–Whitney U test and added to the plot. Comparisons were performed within each cell type. **e** STRING analysis of the interaction network between proteins that are downregulated in IMP1 siRNA treated neurons, where the cognate encoding transcripts are also bound by IMP1 (as identified in 4a). The analysis reports on in the GO category "microtubule-based process" (PANTHER). Network protein-protein interaction (PPI) enrichment *p* value < 1.0e-16. *P* value was calculated using a hypergeometric test. Source data are provided as a Source Data file.

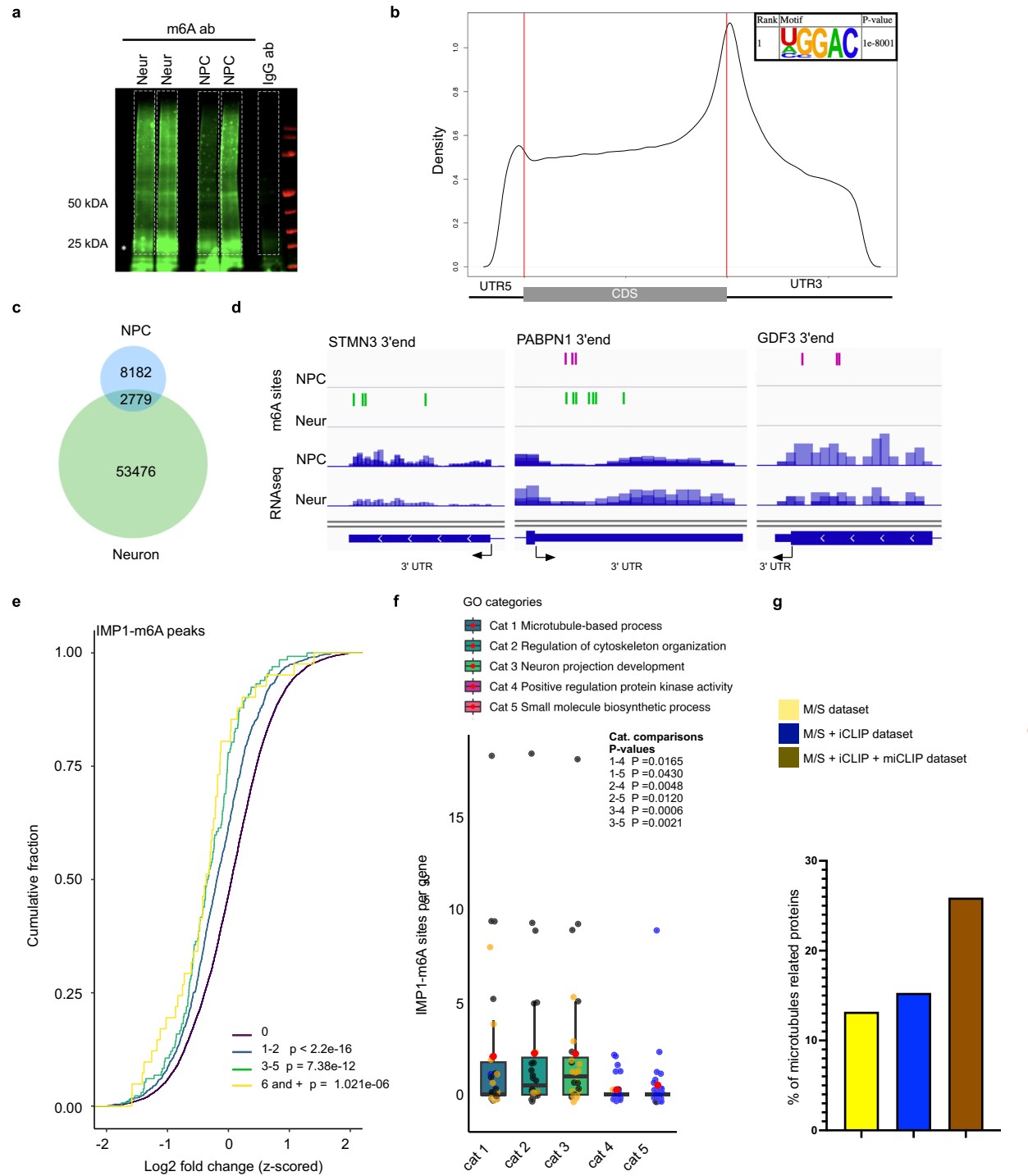

paragraph in the context of methylation. In neuron-specific targets, the mRNAs encoding proteins downregulated in the IMP1 KD experiment have a higher number of IMP1-m6A peaks - where IMP1 peaks overlap with an m6A site - compared to proteins that are not regulated (Supplementary Fig. 5g). Indeed, both at the transcript and peak level we observe that IMP1-m6A overlap is higher in the regulated transcripts (Supplementary Fig. 5h), and the overall increase in IMP1 target expression is directly correlated to the number of m6A-IMP1 sites (Fig. 5e), with large changes observed even for single sites. In further analysis, we linked this increase to functional enrichment of IMP1 regulation, and quantified the number of IMP1-m6A sites for different GO categories. As predicted, targets related to microtubule regulation and neuronal pathways show a significantly higher number of m6A-

IMP1 sites compared to unrelated pathways (Fig. 5f). Consistent with the role that methylation plays in the overall IMP1-mediated regulation and more particularly of the microtubule network, the proportion of IMP1-upregulated proteins that are part of the microtubule network increases as we filter our MS data for (i) IMP1-bound targets and (ii) IMP1-m6A bound targets (Fig. 5g). The role of m6A in IMP1-mediated gene regulation was further explored using a luciferase-based reporter assay in HeLa cells, which are more amenable to genetic manipulation. A portion of MAP2 3′UTR containing an IMP1 peak that overlapped with m6A sites was cloned downstream of *Renilla* luciferase ORF (Fig. 6a, b). We similarly cloned an IMP1-m6A peak in the DCX 3′UTR, to validate our observations. The luciferase activity was significantly reduced when vectors were co-transfected with an siRNA

**Fig. 5 | IMP1 target sites are selectively enriched in m6A methylation for IMP1-regulated genes. a** Representative LI-COR scanning visualization of poly(A) + RNA crosslinked to an m6A antibody or IgG in neurons, immobilized on a nitrocellulose membrane. RNA is visualized via the ligated infrared adaptor. The portion of the membrane excised and used to generate miCLIP libraries is framed in dashed white lines. For both neurons and NPCs, n = 2 technical replicates from 4 independent iPSC lines (biological replicates). NPC = neural precursor cells; Neur = neurons. **b** Metagene plot showing m6A site distribution in neurons. The motif with the most significant p value according to the HOMER analysis is displayed. P value was calculated within the HOMER package using a binomial distribution test. **c** Venn diagram displaying the number of m6A sites in neurons and/or NPCs. **d** m6A sites detected by miCLIP (top) and RNAseq (sequencing reads, bottom) are displayed on the 3′end of three example genes in NPCs and neurons, using Integrative Genomics Viewer (IGV) software. 3′ untranslated region (3′UTR) and last exon are marked by a blue box and separated by an arrow. NPC = neural precursor cells; Neur = neurons. **e** Cumulative distribution plot of the log2 FC in protein expression between IMP1 knockdown and control neurons as detected by MS, was plotted for corresponding RNA classified by the number of IMP1-m6A peaks detected by iCLIP and miCLIP. For MS, n = 3 independent iPSC lines for each condition (biological replicates); for iCLIP, n = 3 technical replicates from 3 independent iPSC lines (biological replicates); for miCLIP, n = 2 technical replicates from 3 independent iPSC lines + n = 1 technical replicate from 1 independent iPSC line (biological replicates). P values

were calculated using two-sided Kolmogorov–Smirnov test comparing 1–2 peaks or 3–5 peaks or 6 and + peaks to 0 peaks. The corresponding P values are reported on the plot. **f** Number of IMP1 peaks on m6A sites (IMP1-m6A peaks) calculated by superimposing the results of the IMP1 iCLIP and miCLIP analyses in different GO categories determined by the PANTHER analysis. The three microtubule/cytoskeleton/neuronal terms and two cytoskeleton-unrelated terms displayed are determined by the PANTHER analysis shown in Fig. 3d Data presented as boxplots where the centre line is the median, limits are the interquartile range and whiskers correspond to 1.5 times the interquartile range. Outliers are not displayed for clarity. Red dots represent the mean. The categories "Cytoskeleton organization", "Microtubule-based process", "Neuron projection development" contain many common genes. Black dots represent transcripts which are present in more than one of these categories, while orange dots represent transcripts in a single category. "Positive regulation protein kinase activity" with "Small molecule biosynthetic process" include different transcripts, represented by blue dots. **g** Barplot showing the percentage of proteins belonging to the "Microtubules-based process" terms as defined by the PANTHER GO classification for each of the following categories: downregulated proteins as defined in Fig. 3a, downregulated proteins where the encoding mRNA is directly bound by IMP1 as defined in Fig. 4a, downregulated proteins where the encoding mRNA is directly bound by IMP1 as defined in Fig. 4a where IMP1 binds to an m6A site as determined by miCLIP. Source data are provided as a Source Data file.

against IMP1, the m6A methyltransferase METTL3 or both. Notably, a concordant trend can be observed for both 3′UTR RNAs, where the effect of IMP1 silencing is larger than that of the silencing of the methyl-transferase. Cotransfection with both siRNAs showed a more pronounced effect than with each siRNA individually. This is consistent with a model whereby IMP1 binding to specific targets can be enhanced by m6A methylation, as we proposed based on the structure and biophysical characterisation of the IMP1 KH4-m6A interaction[24]. Importantly, overexpression of IMP1 leads to increased expression, which is abrogated when m6A is reduced through METTL3 KD (Fig. 6c). The regulatory effect of m6A on IMP1 binding was validated in neurons by assessing the result of METTL3 KD on the regulated microtubules genes (Fig. 6d). Reducing the level of METTL3 resulted in a decrease of three representative targets, including the regulators MAP2 and DCX, further confirming the role of m6A in the regulation of a neuronal network of microtubule protein expression.

## Discussion

Neuronal differentiation is coupled to an upregulation of microtubule factors that allow the assembly of cytoskeletal structures underlying the development of neurites and synapses[25]. An important question in neuronal development is how RNA-binding proteins can control this complex process.

Here, we perform an integrated analysis of the m6A RNA methylome, transcriptome and proteome in human pluripotent stem cells undergoing neurogenesis. These data, together with the RNA binding landscape and the functional output of IMP1, show that this essential factor binds to and regulates a large network of targets including tubulins and microtubule regulatory proteins during the critical transition between neural precursors and terminally differentiated neurons.

The understanding of how IMP1 and similar proteins coordinate morphological changes in developing neurons requires consideration of the system-wide selection of the RNA targets and regulatory action of these proteins. Our iCLIP data indicate that IMP1 binds to a large ensemble of mRNA targets in both NPCs and neurons, but also that these two ensembles are different, i.e. we observe a specialisation of IMP1 to neuronal targets upon differentiation. IMP1 binding mediates regulation at the protein, but not the RNA level, which is different from its established widespread regulation of mRNA stability reported in many studies using highly proliferating cells[7,9,10,17]. IMP1 regulation is dependent on cellular differentiation in the NPC-to-neuron transition,

and our immunofluorescence data indicate it is necessary to build the complexity of the neurons' network of connections. Notably, ACTB abundance was not significantly dysregulated following IMP1 knockdown, which indicates the IMP1 function we discuss here is distinct from the established regulation of beta-actin local translation[1,4,26] and highlights the need for a system wide analysis of protein(s) binding and function at different stages of development.

A key question is then how IMP1 selectively targets neuronal RNAs during differentiation. Our data show a substantial increase in the transcriptome's m6A 'blueprint' during the NPC-to-neuron transition in human neurodevelopment, likely mediated by the upregulation of the METTL3 m6A writer enzyme and downregulation of the ALKBH5 eraser enzyme. This change in methylation increases the number of IMP1 binding peaks in neuronal RNAs, indicating a higher occupancy on the regulated transcripts. Using first a reporter system in HeLa cells (due to experimental tractability for genetic manipulations), and next validating these results in our human neuronal model, we confirm that IMP1-regulated protein expression of microtubule targets is m6A dependent. Notably, we show that IMP1 binding and regulation of the mRNA targets is not accompanied by the upregulation of the IMP1 protein itself but rather by a decrease in the concentration of IMP1. This suggests a mechanism whereby a lower protein concentration and a site-specific increase in affinity mediated by m6A methylation together increase IMP1 selection of neuronal (microtubule) targets (Fig. 6e). In this working model, the protein interaction with individual RNAs depends on the availability of protein and the affinity of individual binding sites. In conditions of limited protein availability, methylation enhances the functional interaction of a set of highly expressed neuronal target mRNAs with the protein. This is consistent with our recent structural and biophysical data showing that IMP1 KH4 directly recognizes m6A methylated RNA via IMP1 KH4 providing a few-fold increase in affinity and an advantage in binding[24]. We propose that, during neuronal differentiation, an m6A regulatory layer directs IMP1 on a subset of its neuron-specific targets in order to promote important morphological changes. Neuronal mRNAs have been reported to be highly enriched in m6A, and several non-canonical m6A readers (e.g. FMRP, hnRNPA2B1) have been reported to play a role in neuronal differentiation. The developmentally-regulated molecular mechanism of IMP1 specialisation that we describe here may represent not only a key principle of IMP1-mediated differentiation, but a more generalizable design principle in the regulation of morphological changes that accompany cell state transitions during development.

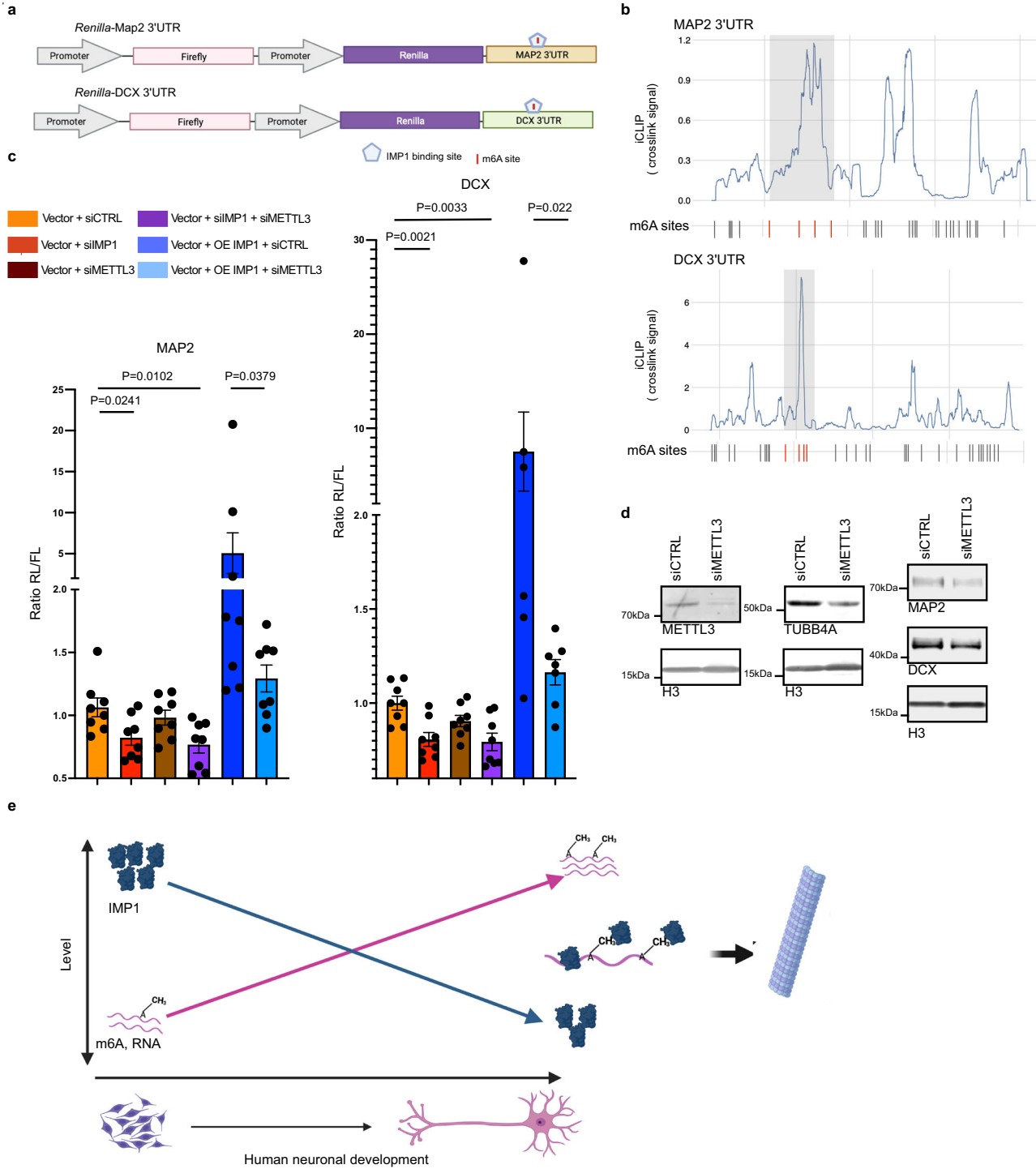

## Methods

### Ethics statement

Experimental protocols were all carried out according to approved regulations and guidelines by UCLH's National Hospital for Neurology and Neurosurgery and UCL's Institute of Neurology joint research ethics committee (09/0272).

### Cell culture

All cell cultures were maintained at 37 °C and 5% CO2. The hiPSC were cultured on Geltrex coated plates with Essential 8 Medium and passaged using 0.5 mM EDTA. hiPSC-derived NPCs and neurons were cultured and differentiated according to previously published protocol[15]. Briefly, hiPSCs were first differentiated by plating to 100%

confluency in medium consisting of DMEM/F12 Glutamax, Neurobasal, L-Glutamine, N2 supplement, non-essential amino acids, B27 supplement, β-mercaptoethanol and insulin. Treatment with small molecules from day 0–7 was as follows: 1 μM Dorsomorphin, 2 μM SB431542, and 3.3 μM CHIR99021. At day 4 and 11, the cell layer was enzymatically dissociated using 1 mg/ml of dispase and plated in one in two onto geltrex coated plates in media containing 1 μM Rock Inhibitor. From day 8 cells were patterned for 7 days with 0.5 μM retinoic acid and 1 μM Purmorphamine. On day 14 cells were treated with 0.1 μM Purmorphamine for a further 4 days to generate spinal cord neuron precursors (NPCs). NPCs were either expanded or terminally differentiated with 0.1 μM Compound E (CE) to promote cell-cycle exit. For all experiments cells were harvested at the NPC stage or mature

**Fig. 6 | m6A methylation modulates the regulatory action of IMP1 on the microtubule-related targets. a** Gene reporter assay constructs used in the study. Constructs were derived from the PsiCheck-2 vector and include the *Renilla* and *Firefly* luciferases ORFs and various 3′UTR sequences with m6A methylated IMP1 binding sites. Image created with BioRender.com released under a Creative Commons Attribution-NonCommercial-NoDerivs 4.0 International license. **b** Mapping of IMP1 iCLIP crosslink signal expressed in counts per millions (top) and m6A sites (dash, bottom) onto the 3′UTR of selected genes in neurons. Merged signal from all replicates is shown for iCLIP. The grey squares highlight the sections of the 3′UTRs used for the Luciferase assay experiments, which contain IMP1 binding sites overlapping with m6A sites. m6A sites located in the cloned regions are highlighted in red. **c** Relative luciferase activity was determined by a dual-luciferase assay system. Rluc-MAP2-UTR3 or Rluc-DCX-UTR3 constructs were co-transfected in HeLa cells with (1) siRNA CTRL, (2) siRNA CTRL + sRNA IMP1, (3) siRNA CTRL + siRNA METTL3, (4) siRNA IMP1 and siRNA METTL3, (5) siRNA CTRL with IMP1 expressing vector, or (6) siRNA METTL3 with IMP1 expressing vector. For MAP2 $n = 8$ replicates (4 independent experiments), for DCX $n = 8$ replicates for conditions 1, 2 and 3 (4 independent experiments), $n = 6$ and 7, respectively for conditions 5 and 6 (3–4 independent experiments). Data is presented as the mean ± SEM. Where data were found to be normally distributed, a two-sided unpaired Student's *t* test was employed; where at least one of the datasets were not normally distributed when comparing 2 conditions, a two-sided Mann–Whitney U test was used. *P* values are displayed on the plot. RL stands for *Renilla* luciferase and FL is for firefly luciferase. **d** Western blot analysis showing expression of IMP1, METTL3, TUBB4A, MAP2, DCX and H3 as loading control in neurons treated with non-targeting control siRNA (siCTRL) or siRNA targeting METTL3 (siMETTL3). Representative images from 2 independent experiments performed on three technical replicates from 2 independent iPSC lines (biological replicates). **e** A working model for m6A regulation of IMP1 binding during development. As IMP1 abundance during differentiation decreases, increased m6A methylation of a subset of targets favours the selective recruitment of the protein. Image created with BioRender.com released under a Creative Commons Attribution-NonCommercial-NoDerivs 4.0 International license. Source data are provided as a Source Data file.

neurons stage (day 7 post CE treatment). Two clones (clone 1 and 4) were made in house[27]. Other clones (2, 3, 5 and 6) are commercially available – clone 2; Coriell, identifier ND41866*C, clone 3; Thermo Fisher, identifier A18945, clone 5; Cedars-Sinai, identifier CS0002iCTR-nxx and clone 6; NIH CRM, identifier CRMi003-A. Four male (C1, C2, C5, C6) and two females clones (C3, C4) were used in this study.

HeLa cells were obtained from Cell Services at the Francis Crick Institute and were cultured in DMEM supplemented with 10% FBS, 2 mM L-glutamine and 1% Penicillin/Streptomycin. Cells were authenticated using STR PCR profiling and were routinely tested for Mycoplasma. No contamination was detected at any point.

### Primary antibodies
Rabbit anti-IMP1 (MBL, RN007P), dilution 1:1000 (WB), dilution 1:100 (IF), 5ug for iCLIP rabbit anti-IgG (Proteintech, 30000-0-AP), 5ug for miCLIP and iCLIP rabbit anti-GAPDH (Cell signalling 14C10), 1:1000 (WB), mouse anti-Actb (Sigma, A2228), 1:1000 (WB), rabbit anti-H3 (Abcam, ab201456), 1:2000 (WB), chicken anti-Homer-1 (Synaptic System, 160006), dilution 1:100 (IF), mouse anti-Synaptotagmin1 (Synaptic System, 105011C3), dilution 1:100 (IF), mouse anti-SMI-35 (BioLegend, 835603), dilution 1:100 (IF), rabbit anti-MAP1B (Proteintech, 21633-1-AP), dilution 1:100 (IF), mouse anti-MAP2 (Abcam, ab11267), 1:1000 (WB), chicken anti-MAP2 (Abcam, ab5392), dilution 1:100 (IF), chicken anti-beta III tubulin (Abcam, ab41489), dilution 1:100 (IF), mouse anti-beta IV tubulin (Abcam, ab11315), dilution 1:100 (IF), rabbit anti-DCX (Proteintech, 13925-1-AP), 1:1000 (WB), 1:100 (IF), mouse anti-MAPT (Abcam, ab80579), dilution 1:100 (IF), rabbit anti-CRIPT (Proteintech,11211-1-AP), dilution 1:100 (IF), mouse anti-m6A (Abcam, ab151230), 5ug for miCLIP, rabbit anti-METTL3 (Proteintech, 15073-1-AP), 1:1000 (WB).

### siRNA knockdown
For RNAseq and proteomics experiments NPCs were plated in 12-well (RNAseq) or 6-well plates (proteomics) in N2B27 media at a density of $2.5 \times 10^5$ cells/well or $2.5 \times 10^6$ cells/well respectively. They were transfected the next day with siRNA directed against IMP1/IGF2BP1 or non-targeting siRNAs as negative control. A concentration of 30 pmol for 12-well plate and 300 pmol for 6-well plate for IMP1 was used. For m6A validation effect on protein expression, NPCs were plated in 12-well in N2B27 media at a density of $2.5 \times 10^5$ cells/well. Cells were transfected the next day with 10 pmol of siRNA directed against METTL3 or non-targeting siRNAs as negative control. Lipofectamine RNAiMax was used as a transfection reagent according to the manufacturer's instructions. After overnight incubation, the media was changed to FGF in N2B27 to maintain cells at the NPC stage or 0.1 μM Compound E in N2B27 to allow terminal differentiation to neurons. Samples were harvested for either protein or RNA extraction at 6 days after media change. Knockdown efficiency was systematically assessed by WB.

### Immunofluorescence staining
Cells were fixed in 4% paraformaldehyde in PBS for 10 min at room temperature. For permeabilization and non-specific antibody blocking, 0.3% Triton-X containing 5% bovine serum albumin (BSA) (Sigma) in PBS was added for 60 min. Primary antibodies were made up in 5% BSA and then applied overnight at 4 °C. After three washes with PBS, species- specific Alexa Fluor-conjugated secondary antibody at 1:1000 dilution in 5% PBS-BSA was applied in the dark for 60 min. Cells were washed once in PBS containing Dapi, 4′,6- diamidino-2-phenylindole nuclear stain (1:1000) for 10 min. Images were taken using the Zeiss 880 inverted confocal microscope or the VT-iSIM.

### Western blotting
Cells were washed with cold PBS 1X on ice and sonicated in RIPA buffer (25 mM Tris-HCl pH 7.6, 150 mM NaCl, 1% NP-40, 1% sodium deoxycholate, 0.1% SDS) supplemented with 1X Protease Inhibitor Cocktail. Supernatants were cleared of debris by 15 min centrifugation at $16,000 \times g$ at 4 °C. Protein quantification was performed using the Pierce BCA protein assay kit according to the manufacturer's instructions. Equal quantities of proteins were supplemented with 4x Nupage loading buffer containing 1 mM DTT and incubated at 90 °C for 10 min. Samples were separated onto 4−12% Bis−Tris protein gels in 1X MES buffer and transferred onto a nitrocellulose membrane for 1 h at constant 30 V at 4 °C using wet transfer. Membranes were blocked by incubation in 5% milk in PBS for IMP1 staining, for 1 h at room temperature under agitation, or in 5% milk in PBS-0.05% Tween-20 (PBS-T) for all other staining. Membranes were then incubated with primary antibodies ON at 4 °C in 1% milk in PBS for anti-IMP1 antibody or in 2.5% milk in PBS-T for all other antibodies either 1 h at RT or ON at 4 °C. Membranes were extensively washed in PBS-T and then incubated with LI-COR species-specific secondary antibodies (IRdye680 1:15000, IRdye800 1:15000) at room temperature for 1 h. Unbound secondary antibody was washed in PBS-T 1X three times. Blots were imaged by Odyssey scanning (LI-COR) and quantified using Fiji software[28].

### RNA extraction for sequencing
IMP1 and control siRNA treated NPCs and neurons were washed with PBS 1X and harvested by centrifugation. Total RNA was extracted using Maxwell RSC simplyRNA cells kit including DNase treatment in the Maxwell RSC instrument following manufacturer's instructions. RNA concentration and the 260/280 ratio were assessed using Nanodrop, and the Agilent 2100 alyser was used to assess quality. RNA integrity (RIN) scores were used to quality check samples.

## RNA extraction for qPCR

NPCs and neurons were washed with PBS 1X and harvested by centrifugation. Total RNA was extracted using Maxwell RSC simplyRNA cells kit including DNase treatment in the Maxwell RSC instrument following manufacturer's instructions. RNA concentration and the 260/280 ratio were assessed using Nanodrop.

## RNA extraction for miCLIP

NPCs and neuron samples were washed with PBS 1X on ice. Content of 80–100% of two 6-well plates was lysed in 1.5 mL TRIzol reagent and total RNA was extracted using manufacturer's instruction (all volumes were scaled up according to the initial volume of TRIzol). RNA was resuspended in 30 µl of RNase free water. RNA concentration and the 260/280 ratio were assessed using Nanodrop, and the Agilent 2400 Bioanalyser was used to assess quality. RNA integrity (RIN) scores were used to quality check samples.

## Reverse transcription and qPCR

RevertAid First Strand cDNA Synthesis kit was used to synthesise cDNA using 1 µg of total RNA and random hexamers. Appropriate dilution of the cDNA was then used in qPCR reactions containing PowerUp SYBR Green Master Mix and primer pairs, using the QuantStudio 6 Flex Real-Time PCR System. Specific amplification was determined by melt curve analysis. Gene expression levels were measured using the ΔΔCT method. IMP1 was amplified using the following pair of primers IMP1-FWD 5'-CAGGGCCGAGCAGGAAATAA – 3, IMP1-REV 5'-CAGGGATCAGG TGAGACTGC −3' and normalised on GAPDH gene amplified with the following pair of primers: GAPDH-FWD 5'-ATGACATCAAGAAGGTGG TG- 3', GAPDH-REV 5'-CATACCAGGAAATGAGCTTG-3'.

For the luciferase assay cloning vectors, 200 ng of RNA from neuronal samples was reverse transcribed with superscript III reverse transcriptase (Thermo-Fisher Scientific) and oligo(dT) using the manufacturer's instructions.

## PolyA enrichment

Total RNA was treated with TURBO DNA-free kit using rigorous DNAse treatment conditions according to manufacturer's instructions (Thermo-Fischer Scientific). DNAse free Poly(A) + RNA was prepared using oligo (dT) dynabeads and the following protocol: 1 mg of dynabeads were washed with binding buffer and resuspended in 100 µl of binding buffer (20 mM Tris-HCl, pH 7.5, 1.0 M LiCl, 2 mM EDTA). Volume of RNA was adjusted to 100 µl in RNAse free water and mixed with 100 µl of binding buffer. Samples were incubated at 65 °C for 2 min and immediately put on ice. RNA was thoroughly mixed with washed beads and tubes were rotating head over tail for 5 min at RT. Two washes with wash buffer A (10 mM Tris-HCl, pH 7.5, 0.15 M LiCl, 1 mM EDTA 10 mM Tris-HCl, pH 7.5) were performed followed by elution with 50 µl of elution buffer (20 mM Tris-HCl, pH 7.5, 1 mM EDTA). Samples were then incubated at 80 °C for 2 min under gentle agitation and placed on a magnetic rack. Supernatant was reused for another round of purification after beads were washed with 100 µl of elution buffer and 200 µl of wash buffer. RNA concentration was assessed using Nanodrop.

## Luciferase reporter assays

HeLa cells were plated on 96-well plates (Corning) and allowed to grow overnight. The next day, cells were transfected with the indicated plasmids and siRNAs using lipofectamine 2000 (Thermo-Fisher Scientific). Transfection was done with 50 ng of total plasmid (50 ng when only one plasmid was transfected or two times 25 ng for conditions with two plasmids), together with 2 pmol of total siRNA (2 pmol when only one siRNA was transfected or two times 1pmol for conditions with two siRNA) diluted in OptiMEM with 25 ng of Lipofectamine also diluted in OptiMEM. After 48 h of transfection, old media was removed and 75ul of fresh media per well was added. Luminescence of Firefly

and *Renilla* luciferase were sequentially measured with a Dual-Glo® Luciferase Reporter Assay System (Promega) following the manufacturer's instruction using Ensight Plate Multimode Reader with default setting for luminescence measurement. For each condition, *Renilla* luciferase luminescence reads were divided by the corresponding Firefly luminescence reads. This ratio was then normalised to the average value from siRNA control conditions for each plate. For each condition at least three independent experiments in duplicates were performed. Data presented is the mean +/- SEM. Statistical analysis was performed using the two-sided unpaired Student's *t* test and the two-sided Mann–Whitney U test.

## Plasmid constructs

For luciferase assay, the dual luciferase reporter psiCHECK-2 vector (Promega) was used. 3'UTR portions were cloned downstream of Renilla luciferase gene. Vector also independently transcribes a firefly luciferase reporter which allows normalization of Renilla luciferase expression. 3'UTR fragments were generated by PCR amplification using neuronal cDNA as template, Phusion High-Fidelity PCR Master Mix with HF Buffer (NEB) following the manufacturer's instruction and specific primers (see primer list). For each PCR, Tm was calculated using NEB's Tm calculator tool. After gel purification, PCR fragments were cloned into psiCHECK-2 using the restriction enzymes XhoI/NotI or SglI/NOTI (New England Biolabs). Ligations were performed with a quick ligation kit (New England Biolabs) and ligation products were transformed into DH5α competent *E. coli* (Thermo Fisher Scientific). Transformed DH5α were incubated overnight at 37 °C on LB agar plates containing 100 µg/mL ampicillin. Colonies were inoculated overnight at 37 °C in liquid cultures containing 100 µg/mL ampicillin. Plasmids were isolated using the QIAprep Spin Miniprep kit (Qiagen). The presence of the 3'UTRs was assessed using Sanger sequencing (Source Bioscience) and midipreps (Invitrogen™ PureLink™ HiPure Plasmid Filter Midiprep Kit) were performed on clones containing the correct inserts. Human IMP1 ORF cDNA clone expression plasmid was obtained from Stratech (HG18209-UT-SIB).Primers: DCX_UTR3_F 5' AAAAACTCGAGtgcccagctgacaagactaa 3', DCX _UTR3_R: 5' AAAAAGCG GCCGCccatgggtggattttttctcttc, MAP2_UTR3_F: AAAAAGCGATCGCTtt cattaggatggactcgt, MAP2 _UTR3_R: 5' AAAAAGCGGCCGCttttatagcta tagcttccc 3'.

## Immunofluorescence quantification

For immunofluorescence quantification a maximum projection of the images was taken from the Z stack. Using CellProfiler[29] the nuclei were then filtered to remove dead cells. The nuclear mask was expanded by 15 pixels and this region was defined as the cytoplasm. To define the neurites, a mask was created from βIII-tubulin, with the nuclei and cytoplasmic compartments removed. The mean intensities for each compartment were calculated, using the defined masks. For synaptic particle quantification, thresholding was used to define SYT1 particles. The particles were then quantified and the area measured. For the branching analysis, the maximum projection of βIII-tubulin underwent pre-processing to remove noise and binarization and then skeletonised using the Skeletonize3D plugin for ImageJ. The AnalyseSkeleton plugin was then used for branch quantification.

## Individual-nucleotide resolution UV-crosslinking and immunoprecipitation of protein-RNA complexes (iCLIP)

iCLIP was performed as previously described[30] with minor modifications. Briefly, three biological and three technical replicates for NPCs and neurons were cross-linked at 300mJ and then lysed in 1 ml of IP lysis buffer. RNA fragmentation was performed with 0.4 units of RNase I and 2 µl TURBO DNAse added to 1 mL of protein lysate at a concentration of 1 mg/mL. Optimal RNase concentration was previously determined using a concentration gradient: low (0.4U),

medium (0.8U) high (2.5 U) to 1 mg of protein lysate from NPCs and neurons. Optimal IMP1 antibody was previously determined using 1 μg or 5 μg of antibody to 1 mg of protein lysate. To separate protein-RNA complexes, samples were incubated with 5 μg of anti-IMP1 antibody or 5 μg of anti-IgG (negative control) coupled to Protein G beads at 4 °C ON rotating head over tail. RNA was ligated to a pre-adenylated infrared labelled IRL3 adaptor with the following sequence:

/5rApp/AG ATC GGA AGA GCG GTT CAG AAA AAA AAA AAA /iAzideN/AA AAA AAA AAA A/3Bio/. The protein-RNA complexes were then size-separated by SDS-PAGE, blotted onto nitrocellulose and visualised by Odyssey scanning (LI-COR). Desired region (determined from the RNase gradient experiment) was cut from the membrane in small pieces and RNA was released from the membrane by proteinase K digestion and incubation for 60 min at 50 °C. RNA was recovered by Phenol Chloroform extraction. cDNA was synthesised with Superscript IV Reverse Transcriptase. Reverse transcription was performed with primers containing UMIs and barcode (XXXXX) to allow multiplexing: /5Phos/ WWW *XXXXX* NNNN AGATCGGAAGAGCGTCGTGAT /iSp18/ GGATCC /iSp18/ TACTGAACCGC. cDNA molecules were purified using AMPure XP beads, then circularised using Circligase II followed by AMPure XP beads purification. After PCR amplification, libraries were size- selected by gel-purification and size distribution was assessed using Agilent 2100 Bioanalyser. Libraries were quantified by QuBit dsDNA HS Assay. Library composed of the same quantity of cDNA for each sample was sequenced as single-end 100 bp reads on Illumina HiSeq 4000.

### m6A Individual-nucleotide resolution UV-crosslinking and immunoprecipitation (miCLIP)

miCLIP was performed as previously described[31] with minor modifications. Briefly, PolyA RNA (1 μg) was brought to 30 μl volume with nuclease-free water and incubated with 3 μl of 10X Fragmentation Buffer at 60 °C for 15 min. Samples were immediately placed on ice and 3.3 μl of Stop Solution was added. Samples were incubated with 10 μl of 1 μg/μl of anti-m6A antibody in 500 μl IP buffer (50 mM Tris-HCL, pH 7.4, 100 mM NaCl, 0.05% NP-40) for 2 h at 4 °C rotating head over tail. RNA-anti m6A antibody complexes were crosslinked twice in a 6-well plate on ice with 0.3 J/cm-2 UV light (254 nm) in a stratalinker. Negative controls (non-crosslink and anti-IgG antibodies) were also prepared. Crosslinked RNA-anti-m6A complexes were mixed with 30 μl of washed Protein G Dynabeads and incubated at 4 °C for 2 h rotating head over tail.

From this point, the library preparation steps were performed as described in the iCLIP section. The correct region from the nitrocellulose membrane was cut according to the previously published protocol[32]. Libraries were sequenced with a single- end 100 bp run using an Illumina Hiseq4000 platform.

### Sample preparation for proteomics analysis

Neurons and NPCs pellets were lysed in 150 μL buffer containing 1% sodium deoxycholate (SDC), 100 mM triethylammonium bicarbonate (TEAB), 10% isopropanol, 50 mM NaCl and Halt protease and phosphatase inhibitor cocktail (100X) on ice, assisted with probe sonication, followed by heating at 90 °C for 5 min and re-sonication. Protein concentration was measured using the Quick Start Bradford Protein Assay according to manufacturer's instructions. Protein aliquots of 50 μg were reduced with 5 mM tris-2-carboxyethyl phosphine (TCEP) for 1 h at 60 °C and alkylated with 10 mM iodoacetamide (IAA) for 30 min in the dark, followed by overnight digestion with trypsin at final concentration 75 ng/μL (Pierce). Peptides were labelled with the TMT 10plex reagents according to manufacturer's instructions. The mixture was acidified with 1% formic acid and the precipitated SDC was removed by centrifugation. Supernatant was then dried with a centrifugal vacuum concentrator.

### High-pH reversed-phase peptide fractionation of TMT labelled peptides

Peptides were fractionated with high-pH Reversed-Phase (RP) chromatography with the XBridge C18 column (2.1 × 150 mm, 3.5 μm, Waters) on a Dionex UltiMate 3000 HPLC system. Mobile phase A was 0.1% (v/v) ammonium hydroxide and mobile phase B was acetonitrile, 0.1% (v/v) ammonium hydroxide. The TMT labelled peptides were fractionated at a flow rate of 0.2 mL/min using the following gradient: 5 min at 5% B, for 35 min gradient to 35% B, gradient to 80% B for 5 min, isocratic for 5 min and re-equilibration to 5% B. Fractions were collected every 42 s, combined orthogonally in 12 fractions and vacuum dried.

### LC-MS analysis

LC-MS analysis was performed on a Dionex UltiMate 3000 UHPLC system coupled with the Orbitrap Lumos Mass Spectrometer. Peptides were loaded onto the Acclaim PepMap 100, 100 μm × 2 cm C18, 5 μm, trapping column at 10 μL/min flow rate and analysed with the EASY-Spray C18 capillary column (75 μm × 50 cm, 2 μm) at 50 °C. Mobile phase A was 0.1% formic acid and mobile phase B was 80% acetonitrile, 0.1% formic acid. For the TMT peptides, the gradient method included: 150 min gradient 5–38% B, 10 min up to 95% B, 5 min isocratic at 95% B, re-equilibration to 5% B in 5 min and 10 min isocratic at 5% B at flow rate 300 nL/min. Survey scans were acquired in the range of 375–1500 m/z with mass resolution of 120 k, AGC $4 \times 10^5$ and max injection time (IT) 50 ms. Precursors were selected with the top speed mode in cycles of 3 sec and isolated for HCD fragmentation with quadrupole isolation width 0.7 Th. Collision energy was 38% with AGC $1 \times 10^5$ and max IT 86 ms. Targeted precursors were dynamically excluded for further fragmentation for 45 s with 7 ppm mass tolerance.

### Database search and protein quantification

The mass spectra were analysed in Proteome Discoverer 2.4 with the SequestHT search engine for protein identification and quantification. The precursor and fragment ion mass tolerances were set at 20 ppm and 0.02 Da respectively. Spectra were searched for fully tryptic peptides with maximum 2 missed-cleavages. TMT 6plex at N-terminus/K and Carbamidomethyl at C were selected as static modifications and Oxidation of M and Deamidation of N/Q were selected as dynamic modifications. Peptide confidence was estimated with the Percolator node and peptides were filtered at $q$ value < 0.01 based on decoy database search. All spectra were searched against reviewed UniProt Homo Sapiens protein entries. The reporter ion quantifier node included a TMT 10plex quantification method with an integration window tolerance of 15 ppm. Only unique peptides were used for quantification, considering protein groups for peptide uniqueness. Only peptides with average reporter signal-to-noise>3 were used for protein quantification. Proteins were normalised based on total peptide signal-to-noise per sample and scaled to mean across samples (mean=100). Statistical analysis was performed in Perseus 1.6. Log2-ratios of knockdowns versus matched controls were computed followed by column z-score transformation. Differentially expressed proteins were defined by a two-sided one-sample $t$ test using the different clones as replicates. Proteins with log2 FC (z-scored) < −1 or > 1 with $p$ value < 0.05 were considered as differentially expressed.

### RNA-sequencing data and gene expression analysis

Poly(A)+ selected reverse stranded RNA sequencing libraries were prepared from total RNA using KAPA mRNA HyperPrep Library kit from Illumina®, with 200 ng of total RNA as input. Libraries were sequenced on a HiSeq 4000 platform. A total of 30 million of 100 bp-long paired-end strand-specific reads were sequenced per sample. Raw mRNA sequencing reads were trimmed using TrimGalore v0.6.5 (https://www.bioinformatics.babraham.ac.uk/projects/trim_galore), quality evaluated using FastQC 0.11.7, (https://www.bioinformatics.babraham.ac.uk/projects/fastqc/) and aligned to GRCh38.p13 human

genome using splice-aware aligner STAR v2.6.1[33] with default parameters. Aligned reads were quantified using --quantMode (TranscriptomeSAM GeneCounts) function. All samples had a percentage of uniquely map reads over 90%.

Differential gene expression was measured using DESeq2[34] with STAR count matrix as an input (DESeqDataSetFromMatrix function). Results were generated by comparing siRNA control to siRNA IMP1 transfected cells in neurons or NPCs. A gene was considered significantly differentially expressed if p value < 0.05 and log2 FC < −1.5 for downregulated genes and >1.5 for upregulated genes. To remove background, genes with zero counts were excluded. For comparing gene expression level between NPCs and neurons we reanalysed previously generated RNA-seq data from the lab[15] - GSE98290. Fastq files from control samples at day 14 (NPC) and day 21 (neurons) of differentiation (SRR5483813, SRR5483814, SRR5483799, SRR5483815, SRR5483812, SRR5483797, SRR5483798) were aligned to GRCh38.p13 human genome using STAR v2.6.1. and using gencode.v36.annotation.gtf file (--sjdbGTFfile). Aligned reads were quantified using --quantMode (TranscriptomeSAM GeneCounts) function. Bedgraph files were generated using deepTools bamcoverage package[35] with default parameters.

### Processing of iCLIP data

iCLIP reads from 9 neuronal and 9 NPCs samples and IgG control were processed according to iCLIP analysis methods using the iMaps webserver (https://imaps.genialis.com/iclip) for standardised primary analysis. iMaps is based on the icount package[36]. Briefly, the following steps were performed: demultiplexing using experimental barcodes, UMI identification, adapter trimming, STAR pre-mapping to rRNAs and tRNAs, STAR alignment to genome, crosslink site assignment, duplicate removal by UMI sequence, peak calling (paraclu), generation of summary files based on crosslink events on gene type, biotype, or gene region. Significant crosslink sites were defined using the 'iCount peaks' tool on the iMaps web server while peaks were defined by clustering the significant crosslink sites using default parameters. Control quality checks (FastQC 0.11.7, https://www.bioinformatics.babraham.ac.uk/projects/fastqc/, PCR duplication ratio, quality of sequencing and alignment statistics from iMaps) were performed on each individual sample. All samples showed high quality with low PCR duplicate ratio (1.8 to 6.1). All samples had a similar number of uniquely mapped reads (on average 7. 3E + 06 for neurons and 5.7E + 06 for NPC) except for two NPC samples which were further excluded from the analysis. Crosslink or peak bed files from biological and technical replicates for neurons (9) and NPCs (7) samples were merged using iMaps group function. Peaks were then called and the output bed or bedgraph files were used for further analysis. For all samples the human GRCh38 genome build and GENCODE version 36 annotation were used. For comparison between neurons and NPC samples and to account for gene expression, normalisation at gene level was performed using Deseq2 by inputting the number of IMP1 crosslink count per gene obtained from iMaps and using gene count values obtained from RNAseq experiment (GSE98290, see "RNA-sequencing data and gene expression analysis" section) as a covariate. When required, normalisation for library size was performed within the DESeq2 analysis or *clipplotR* (see visualisation of iCLIP and miCLIP data method section). To obtain IMP1 expression level independent binding sites, peaks from neurons (9) and NPCs (7) samples were clustered together using the iCount clusters option from iMAPS with a window of 20 nucleotides. Resulting bed file was used as a reference and each individual sample coverage over these peaks was calculated using a bedtools map v2.30.0. Values from each binding site in neurons and NPCs were compared using DESeq2 and gene count values as a covariate. Genes with less than 10 cDNA (iCLIP or RNAseq) in 5 samples were discarded using Deseq2 rowSums function. P values were calculated using an LRT test. A threshold of 1 <Log2 Fold Change < −1 and adjusted p value < 0.05 was used to determine expression independent binding sites between NPCs and neurons. PCA plots were generated using the number of crosslink counts per gene obtained from iMaps.

### miCLIP analysis and m6A site calling

miCLIP reads from 8 neuronal and 8 NPC samples and non-crosslinked control were processed according to iCLIP analysis methods using the iMaps web server for standardised primary analysis (See "Processing of iCLIP data"). Significant crosslink sites were defined using the 'iCount peaks' tool on the iMaps web server while peaks were defined by clustering the significant crosslink sites using default parameters. Control quality check (FastQC report, PCR duplication ratio, quality of sequencing and alignment statistics) was performed on each individual sample. Two samples were discarded based on unique counts number and PCR duplicate ratios. The other samples showed high quality with low PCR duplicate ratio (1.9–15). Correlation between replicates was assessed using deepTools multibamSummary function. Default parameters were used except for binSize (50,000 bases was used). Heatmap was then plotted with deepTools plotCorrelation function using Pearson method. Mutation (CIMS) and Truncation (CITS) site calling was performed as previously described[31,37]. Briefly, low-quality bases and adaptor sequences were all removed using FLEXBAR tool (-f i1.8 -as AGATCGGAAGAGCGGTTCAG --pre-trim-phred 30 -s -t sample). Reads were demultiplexed based on 5′ barcodes for individual replicates using pyCRAC[38]. Reads were processed in pooled or separate replicate modes using the CTK package (https://zhanglab.c2b2.columbia.edu/index.php/CTK_Documentation).

In brief the barcode was stripped and added to the name of the read. Biological and technical replicates were concatenated (CIMS analysis). PCR duplicates were collapsed using pyFastqDuplicateRemover.py script from pyCRAC. The header was transformed to be compatible with CTK analysis (mawk -F '[_/]' '/^>/{print $1"_"$2"_"$3"/ "$4"#"$3"#"$2; getline($9); print $9}'). Reads were aligned to the human genome (hg38) using bwa (aln -t 4 -n 0.06 -q 20). Positions of C → T from mapped reads were obtained using the CIMS software package[39]. Briefly, aligned reads were parsed using parseAlignment.pl to generate a bed file of read coordinates and mutation coordinates. PCR duplicates were collapsed based on read coordinated and barcode identities using tag2collapse.pl script. To get the mutations in unique tags joinWrapper.py script was used. C to T transitions were extracted and a bed file generated. Mutation reproducibility was evaluated using CIMS.pl script. For each mismatch position, the unique tag coverage ($k$) and the number of C → T transitions ($m$) were determined. C → T transitions located in DRACH motifs were called using kmer.annotate.cims.sh script based on ($m \geq 2$ or $m \geq 5$) and transition frequency ($1\% \geq m/k \leq 50\%$)[31]. Position of truncations from mapped reads were obtained using the CITS software package[37]. Briefly, after the mutation removal step from PCR duplicates in the CIMS strategy, deletions were obtained from the mutation file using getMutationType.pl on each individual sample. Truncations events were then identified using CITS.pl scripts[31]. Outputs with P < 0.05 or *P* < 0.001 were used for further analysis. Technical and biological replicates were then concatenated. CITS and CIMS final bed files were merged using bedtools mergeBed function.

### Analysis of enrichment of IMP1 peaks and IMP1-m6A peaks

IMP1 and m6A bed files were annotated using bedtools mergeBed package[40] and by intersecting them with gencode.v36.annotation.gtf file downloaded from https://www.gencodegenes.org/human/release_36.html and converted into bed file using BEDOPS[41](gtf2bed). Annotated IMP1 and m6A bed files were filtered for protein-coding genes and processed to produce a table of IMP1 or m6A counts per gene. To obtain the number of IMP1-m6A peaks per gene, an overlap between IMP1 and m6A bed files was performed using the bedtools

intersect package. The resulting bed file was filtered for protein-coding genes and was processed to produce a table of count per gene of m6A sites bound by IMP1. The protein expression data was obtained from the proteomics dataset. The enrichment score was obtained from the Log2ratios of knockdowns versus matched controls followed by column z-score transformation. A pseudocount of 0.001 was added to the expression values to avoid division with 0. The cumulative distribution function was calculated for genes grouped by IMP1-m6A or IMP1 peaks and plotted using R (ecdf package).

## Quantification of microtubules enrichment terms
Percentage of proteins belonging to the "microtubules-based process" term" was calculated as follows: PANTHER GO analysis was performed on downregulated proteins from the MS experiment, or downregulated proteins from MS experiment containing IMP1 peaks in the corresponding RNA as defined by iCLIP, or downregulated proteins from MS experiment with m6A-IMP1 peaks in the corresponding RNA as defined by iCLIP and miCLIP. The percentage of proteins belonging to the term "Microtubules-based process" was quantified based on the total number of proteins in each corresponding dataset.

## Motif analysis
De novo motif enrichment was performed on peak bed files using HOMER (Hypergeometric Optimization of Motif EnRichment) findMotifsGenome.pl package for (i) all unique IMP1 peaks, (ii) unique IMP1 peaks for downregulated proteins where log2 FC < −1, $p$ value < 0.05 from M/S data with the following parameters changed: search for RNA motifs (-rna), window of 10 or 30 base pair (-size 10 or 30) and a length of 5 (-rna -len 5), and (iii) IMP1-m6A bound peaks with the following parameters changed: search for RNA motifs (-rna) and a length of 5 (-rna -len 5). Default settings were used for the other parameters (http://homer.ucsd.edu/homer/motif/rnaMotifs.html).

## Gene Ontology
GO analysis was performed using protein analysis through evolutionary relationships (PANTHER) version 16.0 (http://geneontology.org) for the biological process category, with the organism set to Homo sapiens. Results were sorted hierarchically using the default parameters (test type: Fisher's Exact, correction: FDR). When necessary, redundant GO terms were removed using REVIGO[42] (Reduce & Visualize Gene Ontology) allowing medium GO term similarity.

## Protein-protein interaction network analysis
Interactions between proteins were investigated using the STRING database (Search Tool for the Retrieval of Interacting Genes/Proteins). Default parameters were used.

## Visualisation of iCLIP and miCLIP data
CLIP data were visualised in a comparative manner on individual transcripts using the software *clipplotr* (https://github.com/ulelab/clipplotr). iCLIP signals were normalised on library size and scaled to crosslinks per million. Rolling, which means smoothing with a sliding window of 100 nucleotides, was used. When specified in the legend Integrative Genomics Viewer (IGV) software was used.

## Generation of nervous system m6A map (meta-analysis)
To generate a map of previously detected m6A sites in nervous tissue, bed files from published miCLIP and MeRIP-seq datasets were downloaded from Gene Expression Omnibus (GEO) repository - GSE106607, GSE106423, GSE71154, GSE29714, GSE37005 or obtained directly from publication (supplementary data) Yu et al. [43]. When necessary, genome coordinates were converted to hg38 genome assembly using Lift Genome Annotations (https://genome.ucsc.edu/cgi-bin/hgLiftOver). Duplicated reads were first removed.

When a dataset contained information from different conditions, it was split into corresponding bed files based on sample ID. Bed files related to the nervous system (brain and neurons) were then merged using the merge command line from bedtools. with default parameters. Overlap between the nervous dataset and neuron miCLIP bed files was performed using bedtools intersect package with default parameters.

## Metagene plots
Metagene plots were generated from CITS and CIMS bed files using MetaPlotR pipeline as previously described[44]. For the comparison between NPC and neurons scaling to 5′ and 3′UTR lengths step has been omitted.

## Statistical analysis
Statistical analyses were performed using RStudio (v. 4.1.2), or GraphPad Prism 7. Normality was assessed using the D'Agostino-Pearson or Kolmogorov-Smirnov tests. For analyses where some, or all, datasets failed a normality test, a non-parametric analysis was used. Paired or unpaired two-sided Student's $t$ test was performed when comparing two categories, except where stated otherwise. When more than two groups were compared, one-way ANOVA was used, except where stated otherwise. For cumulative distribution, a two-sided Kolmogorov–Smirnov test was used to calculate $P$ values. $P$ values less than 0.05 were considered to be significant. Graphs and error bars reflect means ± SEM., except where stated otherwise. Information regarding statistical tests, how significance was determined, and replicate number is given in figure legends.

## Data analysis
Data analyses were performed using Microsoft Excel v16.6, Graphpad Prism 7 and 10, ImageJ (v 2.3.0), RStudio (v 4.1.2), which were also used to generate graphs. Illustrations were created with BioRender.com released under a Creative Commons Attribution-NonCommercial-NoDerivs 4.0 International license.

## Reporting summary
Further information on research design is available in the Nature Portfolio Reporting Summary linked to this article.

# Data availability
The data supporting the findings of this study are available from the corresponding authors upon request. The RNA-seq, iCLIP and miCLIP data generated for this paper have been deposited in NCBI Gene Expression Omnibus (GEO) under accession number GSE203571. RNA-seq data used to estimate the mRNA expression level in NPCs and neurons were obtained from GEO under accession number GSE98290. miCLIP and MeRIP-seq to generate nervous tissue m6A map (meta-analysis) were from GEO under accession number GSE106607, GSE106423, GSE71154, GSE29714, GSE37005 and dataset from Yu et al.[43]. The mass spectrometry raw data have been deposited to PRIDE accession number PXD034341. Crosslink values and transcript coordinates for metagene plots in neurons and NPCs have been deposited in Figshare [https://doi.org/10.6084/m9.figshare.25595715]. Source data are provided as a Source Data file. Source data are provided with this paper.

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

## Acknowledgements

Thanks are due to Jernej Ule and his team in particular Flora Lee, Rupert Faraway, Charlotte Capitanchik, Klara Kuret for crucial advice and discussion for the design of iCLIP and miCLIP experiments as well as the access to reagents and protocols. We would like to thank Yoh Isogai for useful discussions, help during the design of the project, and for critical reading of the manuscript. We would also like to thank Giancarlo Abis, and other members of the Ramos and Patani labs for experimental support, comments and suggestions on the project and manuscript; members of the Crick Advanced Sequencing and Light Microscopy

Facilities and the Institute Cancer Research (ICR) proteomics core facility for support. P.K. and A.R. are supported by the UK Medical Research Council research grant S000305/1. A.R. is also supported by the UK BBRSC research grant S014438/1. R.P. is supported by an MRC Senior Clinical Fellowship (MR/S006591/1) and a Lister Research Prize Fellowship. R.L. is supported by the Idiap Research Institute. This work was also supported by the Francis Crick Institute, which receives its core funding from Cancer Research UK (FC010110), the UK Medical Research Council (FC010110), and the Wellcome Trust (FC010110). The proteomics work of T.I.R. and J.S.C. was funded by the CRUK Centre grant with reference number C309/A25144. For the purpose of open access, a Creative Commons Attribution (CC BY) has been applied.

## Author contributions

P.K., R.P. and A.R. conceived the project, J.H. performed and analysed immunofluorescence experiments, H.C. and P.K. performed the miCLIP experiment, T.I.R. and J.S.C. performed mass spectrometry analysis, P.K. performed all other experiments with support from M.P.H. for cell culture work and analyses, S.E.S., A.M.C., R.L and P.K. performed bioinformatics analysis, P.K., R.P. and A.R. wrote the manuscript with input from all authors.

## Competing interests

The authors declare no competing interests.
