## [Peer Review File · Nature Communications]

m6a methylation orchestrates IMP1 regulation of microtubules during human neuronal differentiationREVIEWER COMMENTS

Reviewer #1 (Remarks to the Author):

Klein et al. studied the role of IMP1 using different omics analyses. I focused on the proteomics quantitative analysis using TMT 10 plex. The authors used well established pipelines: TMT 10-plex LC-MS/MS via Orbitrap Lumos MS coupled with UHPLC system, protein (peptide) identification via PD 2.4 coupled with Percolator at FDR threshold of 1%, reporter ion normalization and quantification, and statistical detection of differentially expressed proteins both by p-value and FDs (z scores). The authors only used unique peptides to infer protein quantities, circumventing the complication that might have arisen otherwise. While this may drop the sensitivity in identification, it may raise the accuracy in quantification, thus making the analysis more robust. I do not see much of flaw in this protein quantification pipeline and would say the conclusion on differentially expressed proteins is solid. The authors already deposited the dataset in a public repository (PRIDE).

Reviewer #2 (Remarks to the Author):

In the manuscript titled "m6a methylation orchestrates IMP1 regulation of microtubules during human neuronal differentiation", the authors used multiple high-throughput methods to characterize how IMP1 regulates mRNA and protein expression using a human induced pluripotent stem cell (hiPSC) neuron differentiation model. The biological question is important but it is disappointing that the manuscript feels like an incomplete work, as the authors did not address the interplay between m6a methylation and IMP1 binding/regulation at all. Furthermore, the results section of CLIP, RNA-seq and mass spectrometry data were largely descriptive and selectively presented, which was insufficient to give the reader a clear picture of how IMP1 binds and regulates gene expression during neurogenesis. Lastly, the clarity and accuracy of the presentation also needs to be improved. Overall, the manuscript needs very substantial revision to be considered for publication in Nature Communications.

The essential revisions for the current manuscript are listed below. However, the authors really need to extend the study to fully establish the interaction between m6a methylation and IMP1 binding/regulation for the manuscript to provide sufficient advance to justify publication in Nature Communications. For example, IMP1 is known to bind CA-rich motifs (Conway et al. Cell Rep. 2015), which overlaps with the DRACH consensus sequence for m6A deposition (Flamand et al. Annu Rev Biochem. 2023). Therefore, the authors need to compare the overlap between IMP1-iCLIP and miCLIP peaks on all and selected targets, both at global and sequence levels. Next, the authors can pick model genes and use reporters to assay whether the IMP1 binding sites and m6A sites individually and/or collectively recapitulate the observed regulatory effects at mRNA and/or protein levels. The authors also need to address whether the IMP1 binding and m6A modification is working synergistically or competitively using reporters and/or endogenous genes. The IMP1-m6a structural/biophysical data mentioned in Discussion will strengthen the impact of the manuscript if included, but it is understandable that the authors plan to reserve them for a separate manuscript.

##Texts. (**: essential revisions)

** -page 2, last paragraph, "IMP1 crosslink sites were...highly-proliferative and cancer cells.": The term "highly proliferative cells" is correct but ambiguous. It will be better to fully acknowledge that IMP1 binding and motif analyses using eCLIP and RBMS have been done previously in hPSCs (reference 15) but hasn't been explored in NPCs and differentiated neurons in the introduction section.

** -page 3, paragraph 2, "To address this, we compared IMP1-RNA binding iCLIP tracks of individual mRNAs, where changes during differentiation can be directly visualised.": The sentence about using a previously generated RNA-seq data to aid this analysis is buried in the Methods section ("For comparing gene expression level between NPC and neurons we reanalysed previously generated RNA-seq data from the lab - GSE98290"). The statement needs to be also included in the main text so the

readers can understand the experiments and analyses correctly.

** -page 3, paragraph 2, "Notably, this asymmetric distribution is exaggerated when considering the peaks of mRNAs encoding proteins involved in axonal development and microtubule organisation, which our Gene Ontology analysis indicates are highly enriched in neuronal IMP1 targets.": What do the authors mean "exaggerated"? More biased toward upregulation? Please clarify and show the results in the main or extended data figure.

** -page 3, paragraph 2, "Finally, and importantly...": In my opinion, "abundance" is more accurate than "concentration" to describe results from qPCR and WB, which do not measure volume/distribution. Please change the wording.

** -page 3, paragraph 2, It is important to stratify targets based on transcript abundance. The authors focused the analysis on the "new" neuronal targets using CLIP and RNAseq data but the analyses are not complete. The authors need to provide a statement in this paragraph to say how many neuron-specific targets (2567 in Extended Data Fig. 2f) are "upregulated" genes (like NEFM?) and "unchanged" genes (like PBX1?) in neurons. Furthermore, after transcript-abundance normalization as shown in Fig. 1g, how many genes have increased binding and how many genes have decreased binding? What are the characteristics of these genes? What are the GO term analyses look like for each group? BTW, Fig. 1f did not include an example gene with "decreased binding", which should be added if a suitable example can be found. These are important biological information that needs to be provided and will be revisited as the manuscript progresses with the m6A information.

** -page 3, paragraph 4, "In NPCs, a similar number of proteins are up and downregulated upon IMP1 KD, while in neurons twice as many proteins are downregulated compared to upregulated": The authors did not acknowledge the possibility that a weaker IMP-1 KD in NPCs (comparing insets of Fig. 2a and Extended Data Fig. 3a) could contribute to the different results in addition to the differentiation states. The authors need to quantitate and show (either in the main or extended data figure) the average IMP1 protein KD levels in the mass spectrometry experiments shown in Fig. 2a and Extended Data Fig. 3a. IMP1 abundance may be readily available in the existing mass spec data. Alternatively, the authors can quantitate the existing WB results and show the numbers. Next, if the IMP1 KD level is indeed lower in NPCs, the authors need acknowledge the aforementioned possibility that the different KD levels might contribute to the difference shown in Fig. 2b in the text.

** -page 4, "This is evidenced by discordant changes in IMP1 protein and RNA targets upon KD.": The authors need to first compare the entire lists of differentially expressed proteins and RNAs in both NPCs and neurons, divide them into 3 groups (DE in both mRNA and protein; DE in mRNA only; DE in protein only), and present the results in a new figure so the readers can clearly understand the impact of IMP1 KD at both mRNA and protein levels. Next, the authors need limit the lists to IMP1 direct targets (by CLIP data) and show the numbers in each of the 3 groups (DE in both mRNA and protein; DE in mRNA only; DE in protein only).

** -page 5, paragraph 3, "RNA methylation modulates IMP1 selection and regulation of the microtubule targets": The authors did not investigate whether the convergence of IMP1- and m6A-mediated regulation on microtubule networks genes is mechanistically distinct or additive. First, the authors need to compare the overlap between IMP1-iCLIP and miCLIP peaks on all and selected targets, both at global and sequence levels. Next, the authors can pick model genes and use reporters to assay whether the IMP1 binding sites and m6A sites individually and/or collectively recapitulate the observed regulatory effects at mRNA and/or protein levels. Lastly, the authors also need to address whether the IMP1 binding and m6A modification is working synergistically or competitively using reporters and/or endogenous genes.

** -page 6, paragraph 4, "This suggests a mechanism whereby a lower protein concentration and a site-specific increase in affinity mediated by m6A methylation together increase IMP1 selection of neuronal (microtubule) targets (Fig.3j)": No data are presented to support this specific mechanism. This model should be experimentally tested (see above).

-page 6, paragraph 4, "This is consistent with our recent structural and biophysical data showing that IMP1 KH4 directly recognizes m6A methylated RNA by IMP1 KH4 providing a few-fold increase in affinity and an advantage (Nicastro et al., unpublished data)": Is there a plan to include these data in the manuscript?

##Figures. (**: essential revisions)

-Fig. 1:

1) Fig. 1c: Please use a different color scheme with stronger contrast (ideally six distinct colors). With the current color scheme, the less abundant categories such as "intergenic" and "intron" are difficult to see. (They could be absent, but it is difficult to tell as shown.) The authors can find sample color schemes at colorbrewer2.org.

**2) Fig. 1d: Need to explain the two red lines in the legends and add an arrow or other indicators to guide the readers to "the peak adjacent to the stop codon" mentioned in the text.

**3) Fig. 1e: The legends are missing.

**4) Fig. 1f: Please explain/state the following in the legends: The numbers for the Y-axis, the locations for the X-axis (last exon with 3' UTR?), and the categories (upregulated or stable) for each gene.

**5) Fig. 1h: The horizontal line below the asterisk is off. Please correct.

-Fig. 3:

**1) Fig. 3h & 3i: Legends were swapped. Please correct.

##Extended Data Figures. (**: essential revisions)

-Extended Data Fig. 2:

1) Appreciate the inclusion of the results from iCLIP optimization experiments.

**2) Fig. 2c: It looks like NPC#9 had the fewest unique counts but NPC#8 was excluded? Please explain or correct.

-Extended Data Fig. 3:

**1) The legends does not match the figures. Please correct.

2) All panels are a bit too small to read clearly.

-Extended Data Fig. 4:

1) Fig. 4f: Is the circle labeled "Nervous" the meta-analysis dataset? Please clarify in the legend. Maybe "Literature" or "Meta-analysis" would be better labels?

Reviewer #3 (Remarks to the Author):

In this paper, the authors aim to establish that, during the differentiation of human neurons, the RNA-binding protein IMP1 is a central regulator in the expression of mRNAs encoding major players in the microtubular network. They also aim to demonstrate the role of m6A methylation in the choice and selection of IMP1 target sites. This study is based essentially on the computational analysis of omics data obtained from pluripotent stem cell cultures during neurogenesis: IMP1 CLIP, transcriptome, m6A RNA methylome, proteome.

The authors recently established that the reader IMP1 recognizes m6A marks independently of the sequence context (NAR 2023). This study brings functional meaning to this role of m6A reader in the context of neuronal differentiation. Overall, the hypothesis is appealing, the scientific approach is appropriate and the functional implications of this study could be significant for the field. The data obtained and their analysis seem to be of good quality (even if I am not an expert in bioinformatics). Nevertheless, some observations are mainly correlative and would benefit from more in-depth study, most particularly as the authors tackle an innovative concept. In my opinion, two important points are missing or need to be significantly strengthened:

(1) The type of regulation exerted by IMP1 on target mRNAs, which remains speculative and would have benefited from a translactome analysis (by ribosome profiling or ribo-seq) and IMP1 CLIP from polysome fractions.

(2) As things stand, the data do not allow us to establish a direct link between m6A and IMP1. For this, it would be necessary to demonstrate (functionally) that the absence of the m6A mark leads to

the loss of IMP1 binding at the sites concerned and the subsequent deregulation of corresponding protein. In this respect, does IMP1 play a direct/functional role on RNA or is it simply a competitor of m6A reader(s)?

Other comments:

Fig 1b. It gives the wrong idea that IMP1 is only present in cell body in neurons. It would be nice to have a Zoom and show separate colors (in white for DAPI to make it more visible). Same for Sup Fig1B: DAPI staining is unclear.

The authors establish that IMP1 iCLIP peaks are independent from mRNA concentrations. What about cell body vs neurite since IMP1 is involved in mRNA localization? Is there any data available on this?

Fig 1d. Do the authors get a preferential distribution of IMP1 on CA-enriched motifs as previously published?

Fig 1f. IMP1 distribution does not seem to be enriched in the 3'-UTR for these selected genes. Does it vary depending on mRNA? Is the distribution different in some specific groups (e.g. tubulin Network)? It seems to be the case since the authors say (p3) "this asymmetric distribution is exaggerated when considering the peaks of mRNAs encoding proteins involved in axonal development and microtubule organization, which our Gene Ontology analysis indicates are highly enriched in neuronal IMP1 targets." Where is this analysis?

Fig 1h. There is a lot of variability of IMP1 protein expression in neurites. This is not the case in NPCs. One of the measures could well be an outlier (in particular the lowest one). The authors must show the WB in extended data section.

It is unclear to me why the authors focused on "microtubule" GO term rather than "myristylation" based on Fig 2c. Perhaps I missed the rationale behind this choice, even more considering that myristylation participates in cellular distribution of proteins in neurons. Could the authors comment on that?

Is there a significant impact of IMP1 KD on neuron function/phenotype? It looks like neuron morphology is not altered at all (nor bIII tubulin staining) (ED Fig 3c).

Minor remarks:

Fig 1: the legend for "1e" is missing

Fig 3: "d" and "e" are mislabeled

Define in the main text "IMP1-m6A peaks"

We thank the three reviewers for their insightful comments about our manuscript. By addressing these comments, we have now a much improved version of the paper.

Reviewer #1 (Remarks to the Author):

Klein et al. studied the role of IMP1 using different omics analyses. I focused on the proteomics quantitative analysis using TMT 10 plex. The authors used well established pipelines: TMT 10-plex LC-MS/MS via Orbitrap Lumos MS coupled with UHPLC system, protein (peptide) identification via PD 2.4 coupled with Percolator at FDR threshold of 1%, reporter ion normalization and quantification, and statistical detection of differentially expressed proteins both by p-value and FDs (z scores). The authors only used unique peptides to infer protein quantities, circumventing the complication that might have arisen otherwise. While this may drop the sensitivity in identification, it may raise the accuracy in quantification, thus making the analysis more robust. I do not see much of flaw in this protein quantification pipeline and would say the conclusion on differentially expressed proteins is solid. The authors already deposited the dataset in a public repository (PRIDE).

Author response: We thank the reviewer for their work on our manuscript, and for the support.

Reviewer #2 (Remarks to the Author):

In the manuscript titled "m6a methylation orchestrates IMP1 regulation of microtubules during human neuronal differentiation", the authors used multiple high-throughput methods to characterize how IMP1 regulates mRNA and protein expression using a human induced pluripotent stem cell (hiPSC) neuron differentiation model. The biological question is important but it is disappointing that the manuscript feels like an incomplete work, as the authors did not address the interplay between m6a methylation and IMP1 binding/regulation at all. Furthermore, the results section of CLIP, RNA-seq and mass spectrometry data were largely descriptive and selectively presented, which was insufficient to give the reader a clear picture of how IMP1 binds and regulates gene expression during neurogenesis. Lastly, the clarity and accuracy of the presentation also needs to be improved. Overall, the manuscript needs very substantial revision to be considered for publication in Nature Communications.

Author response: We thank the reviewer for their work on our manuscript and we respond to the issues raised below.

The essential revisions for the current manuscript are listed below. However, the authors really need to extend the study to fully establish the interaction between m6a methylation and IMP1 binding/regulation for the manuscript to provide sufficient advance to justify publication in Nature Communications. For example, IMP1 is known to bind CA-rich motifs¹, which overlaps with the DRACH consensus sequence for m6A deposition⁽²⁾. Therefore, the authors need to compare the overlap between IMP1-iCLIP and miCLIP peaks on all and selected targets, both at global and sequence levels. Next, the authors can pick model genes and use reporters to assay whether the IMP1 binding sites and m6A sites individually and/or collectively recapitulate the observed regulatory effects at

mRNA and/or protein levels. The authors also need to address whether the IMP1 binding and m6A modification is working synergistically or competitively using reporters and/or endogenous genes. The IMP1-m6A structural/biophysical data mentioned in Discussion will strengthen the impact of the manuscript if included, but it is understandable that the authors plan to reserve them for a separate manuscript.

Author response: We thank the reviewer for these comments, which we comprehensively address below. Before we detail these revisions, it is important to acknowledge that, in our model, the regulatory role of m6A on IMP1 target selection and function is quite different from the role it plays in the regulation of YTH protein-RNA binding. In our model, only a subset of IMP1 targets are regulated by m6A and m6A regulation is specific to the developmental stage. This model is strongly supported by the structural and biophysical information on IMP1-m6A recognition that is now published in *Nucleic Acids Research*³. We now discuss this study in the m6A results section, to better link the two manuscripts.

‘For example, IMP1 is known to bind CA-rich motifs¹, which overlaps with the DRACH consensus sequence for m6A deposition². Therefore, the authors need to compare the overlap between IMP1-iCLIP and miCLIP peaks on all and selected targets, both at global and sequence levels.’

Comparison at the global level. The reviewer asks us to analyse and interpret the overlap of m6A and IMP1 peaks in a) the whole interactome and b) the subsets of mRNA that are regulated by the IMP1. We now report the percentage of IMP1 and m6A overlapping peaks for all targets, and additionally the targets regulated by IMP1 in our proteome-wide analysis (**Extended Data Fig.5h**). As per our model, the percentage of peaks in terms of IMP1 bound m6A targets over total IMP1 targets is ~8%, reflecting that only a subset of IMP1-RNA interactions are regulated by m6A. Importantly, this increases to ~17% as we filter the set of IMP1 RNA targets that are regulated at the protein level. This is consistent with the link between methylation and the IMP1-mediated regulation of protein abundance, as revealed in our paper. Notably, we also report the percentage of targets where an IMP1 peak coincides with an m6A peak, which indicates the extent to which targets *can* be regulated by m6A. The percentage of IMP1 bound m6A targets over total IMP1 targets is 25% and the IMP1 bound m6A targets regulated at the protein level over total IMP1 targets regulated at the protein level is 36%. Based on this we can infer that m6A regulates a defined subset of functional IMP1-RNA interactions.

Comparison at the sequence level. To address this at the sequence level, we have performed an unbiased Homer analysis with a sequence of 5 bases (KH domains can recognise specifically up to 5 nucleotides). We have used default parameters and, initially, a window of either 10 on each side of the cross-link. Two sequences are identified to be enriched in the smaller 10 nt window. The first, which is both highly enriched and highly represented is ACGGA, which contains a nearly-perfect KH4 consensus sequence (CGGAC) (**Extended Data Fig.3a**). The IMP1 KH4 sequence matches the DRACH-consensus and is therefore difficult to differentiate between IMP1 and m6A sites at the sequence level. Next, we used a broader window (30 nucleotides) to test whether the motifs of the other KH domains of IMP1 were also enriched. The analysis of sequences within a 30 nucleotides window yields multiple, different motifs, which is not uncommon when analysing iCLIP data. However, it is worth mentioning that

the most enriched sequences are those for KH1 ⁴ and KH3 ^{5,6}. We discuss this in the first Results section where we state that: 'We also examined whether neuronal IMP1 binding sites were enriched in the consensus motifs of IMP1 individual KH domains using a HOMER-based approach. Our analyses used a sequence of 5 bases (KH domains can recognise specifically up to 5 nucleotides) and default parameters. An initial analysis performed with a narrow 10-nucleotide window identified a highly enriched KH4 core consensus sequence (GGA) ^{5,6}, while a follow-up analysis with 30-nucleotide-window, instead, returned three motifs. The two highest scoring among these (CCGTT and ACACA) contain the core consensus sequences of KH1 (CC(or G)G) ⁴ and KH3 (CA/ACA) ^{5,6} (**Extended Data Fig.3a**). In NPCs, although the results were less clear-cut, the KH4, KH3 and KH1 motifs were also present (data not shown). Notably, a CA sequence has been previously reported to be enriched in IMP1 target sites in previous computational studies ¹. These results connect IMP1 neuronal target recognition with the current molecular understanding of the protein's target specificity.'

Unfortunately, it has been difficult to pin down a reliable structure of the binding sites that includes the distance between the motifs. Notably, the ensemble of sequences available from the regulated RNA is really too small to perform a reliable HOMER analysis. We have attempted such an analysis for the purpose of answering the reviewer's comment as completely as possible, but the output motifs were, as expected, low confidence. Also, the analysis in NPC output was noisy. Nevertheless, we mention the NPCs analysis in the text for completeness.

##Texts. (**: essential revisions)

** -page 2, last paragraph, "IMP1 crosslink sites were...highly-proliferative and cancer cells.": The term "highly proliferative cells" is correct but ambiguous. It will be better to fully acknowledge that IMP1 binding and motif analyses using eCLIP and RBMS have been done previously in hPSCs (reference 15) but hasn't been explored in NPCs and differentiated neurons in the introduction section.

This has now been clarified in page 2 of the introduction, where we state that 'IMP1 binding and motif analyses has previously been performed, but is limited to highly proliferative cells including in cancer and pluripotent human cells' ¹

** -page 3, paragraph 2, "To address this, we compared IMP1-RNA binding iCLIP tracks of individual mRNAs, where changes during differentiation can be directly visualised.": The sentence about using a previously generated RNA-seq data to aid this analysis is buried in the Methods section ("For comparing gene expression level between NPC and neurons we reanalysed previously generated RNA-seq data from the lab - GSE98290"). The statement needs to be also included in the main text so the readers can understand the experiments and analyses correctly.

This statement has now been included in the text, where we state that 'To address this, we first compared IMP1-RNA binding iCLIP tracks of individual mRNAs, where changes during differentiation can be directly visualised (**Fig.1f and Extended Data Fig.2f**), with gene expression, as obtained from re-analysing RNAseq data that we previously generated (Luisier et al. 2018).'

** -page 3, paragraph 2, "Notably, this asymmetric distribution is exaggerated when considering the peaks of mRNAs encoding proteins involved in axonal development and microtubule organisation, which our Gene Ontology analysis indicates are highly enriched in neuronal IMP1 targets.": What do the authors mean "exaggerated"? More biased toward upregulation? Please clarify and show the results in the main or extended data figure.

Yes, that is indeed what we mean and this is now clarified in the paper where we report the ratios of targets that are higher in neurons vs NPCs after transcript-abundance normalization (3.44). We also compare this number with the ratio obtained when we consider only the subsets defined in the axonal development-related (4.5) and microtubule organisation-related (8.1) types of mRNA. The individual GO terms, which have been grouped to obtain a reasonable (~90) group of targets, are reported in **Extended data Fig. 3b**, and the trend is discussed in the manuscript, where we write that 'Notably, this asymmetric distribution is further accentuated when considering the peaks of mRNAs encoding neuronal proteins and cytoskeleton organisation (**Extended Data Fig.3b**).

** -page 3, paragraph 2, "Finally, and importantly...": In my opinion, "abundance" is more accurate than "concentration" to describe results from qPCR and WB, which do not measure volume/distribution. Please change the wording.

The wording has now been changed as follows: 'Importantly, the increase in IMP1 occupancy of these targets cannot be explained by an increase in the abundance of IMP1, as our data indicate that the level of IMP1 in the cell decreases in the transitions from progenitors to neurons in accordance with its expression decreasing during the development of brain and other tissues ^{7,8} (**Fig.1h, Extended Data Fig. 3d**).

** -page 3, paragraph 2, It is important to stratify targets based on transcript abundance. The authors focused the analysis on the "new" neuronal targets using CLIP and RNAseq data but the analyses are not complete. The authors need to provide a statement in this paragraph to say how many neuron-specific targets (2567 in Extended Data Fig. 2f) are "upregulated" genes (like NEFM?) and "unchanged" genes (like PBX1?) in neurons. Furthermore, after transcript-abundance normalization as shown in Fig. 1g, how many genes have increased binding and how many genes have decreased binding? What are the characteristics of these genes? What are the GO term analyses look like for each group? BTW, Fig. 1f did not include an example gene with "decreased binding", which should be added if a suitable example can be found. These are important biological information that needs to be provided and will be revisited as the manuscript progresses with the m6A information.

We thank the reviewer for this suggestion, which has helped to add further insight to our study. We now add the following analyses:

i) report the numbers of up and down-regulated targets within the neuron specific targets (within the 2567 neurons specific targets, 1218 are upregulated, 722 are downregulated, 627 are unchanged) In the text we now discuss that: 'Indeed, when considering the transcripts that our iCLIP experiments identify as neuronal IMP1 targets (2567), 1218 are upregulated,

722 downregulated and 627 have a stable expression in neurons when compared to progenitors (**Extended Data Fig.2g**).¹

ii) discuss the numbers of targets with increased or decreased binding after transcript-abundance normalisation (1315 have increased binding, 348 have decreased binding) and the GO analysis of those targets (Extended data Fig. 3c). The latter indicates that microtubule terms (in yellow) are enriched in targets with increased but not with decreased binding in neurons, which is consistent with the volcano plots displayed in Extended data Fig. 3b and discussed above. These analyses are discussed in the paper where we state that:

“This showed that, while transcript abundance is one important determinant of IMP1-mRNA binding in neurons, the increase in a large (1315) group of iCLIP peaks is independent of mRNA concentration (**Fig.1g**), while a much smaller group of peaks (348) is decreased. Notably, this asymmetric distribution is further accentuated when considering the peaks of mRNAs encoding neuronal proteins and cytoskeleton organisation (**Extended Data Fig.3b**). This is consistent with the results of the Gene Ontology analysis of the developmentally regulated peaks (**Fig. 1e, Extended data Fig, 3c**), revealing an enrichment of microtubule-related mRNAs that are regulated by IMP1.”

Finally, we have now also included examples of decreased binding in neurons compared to NPCs (Extended data Fig. 2f).

** -page 3, paragraph 4, "In NPCs, a similar number of proteins are up and downregulated upon IMP1 KD, while in neurons twice as many proteins are downregulated compared to upregulated": The authors did not acknowledge the possibility that a weaker IMP-1 KD in NPCs (comparing insets of Fig. 2a and Extended Data Fig. 3a) could contribute to the different results in addition to the differentiation states. The authors need to quantitate and show (either in the main or extended data figure) the average IMP1 protein KD levels in the mass spectrometry experiments shown in Fig. 2a and Extended Data Fig. 3a. IMP1 abundance may be readily available in the existing mass spec data. Alternatively, the authors can quantitate the existing WB results and show the numbers. Next, if the IMP1 KD level is indeed lower in NPCs, the authors need to acknowledge the aforementioned possibility that the different KD levels might contribute to the difference shown in Fig. 2b in the text.

We thank the reviewer for highlighting this possibility. The analysis has been done and the result, reported in **Extended data Fig. 4b**, shows that no significant difference of IMP1 abundance was observed in the NPC IMP1 knock down compared to the neuron IMP1 knock down .

** -page 4, "This is evidenced by discordant changes in IMP1 protein and RNA targets upon KD.": The authors need to first compare the entire lists of differentially expressed proteins and RNAs in both NPCs and neurons, divide them into 3 groups (DE in both mRNA and protein; DE in mRNA only; DE in protein only), and present the results in a new figure so the readers can clearly understand the impact of IMP1 KD at both mRNA and protein levels. Next, the authors need to limit the lists to IMP1 direct targets (by CLIP data) and show the numbers in each of the 3 groups (DE in both mRNA and protein; DE in mRNA only; DE in protein only).

We thank the reviewers for this comment. In the paper, we had focused on the significant IMP1-mediated regulation of a large subset of genes at the protein, but not RNA, level. The more detailed analysis of RNAseq changes, which is reported below, confirms that significant changes are visible only for a very few genes (please see Figure below). This, we feel, is not as important as other data in the network analysis our paper focuses on and could dilute the argument. Because of this, we now comment in the paper that ‘Notably, analysis of RNAseq data in both control and IMP1 KD conditions showed that **only a modest number of genes are significantly up or down-regulated, and that IMP1-mediated regulation of the microtubule network occurs at the protein (rather than mRNA) level (Fig 2d). This is evidenced by discordant changes in IMP1 protein and RNA targets upon KD (Fig. 2d).**’ but do not discuss the new layered data. We are open to insert the data in the Supplementary materials if the reviewers consider it is critical.

** -page 5, paragraph 3, "RNA methylation modulates IMP1 selection and regulation of the microtubule targets": The authors did not investigate whether the convergence of IMP1- and m6A-mediated regulation on microtubule networks genes is mechanistically distinct or additive. First, the authors need to compare the overlap between IMP1-iCLIP and miCLIP peaks on all and selected targets, both at global and sequence levels. Next, the authors can pick model genes and use reporters to assay whether the IMP1 binding sites and m6A sites individually and/or collectively recapitulate the observed regulatory

effects at mRNA and/or protein levels. Lastly, the authors also need to address whether the IMP1 binding and m6A modification is working synergistically or competitively using reporters and/or endogenous genes.

This is an important point that relates to the general request of reviewer 2 to further demonstrate causality and explore the functional relation between IMP1 binding and methylation using reporter assays on model genes, as well as to one of the points made by reviewer 3. We have therefore selected a gene which we have shown in neurons is regulated by both IMP1 and methylation (MAP2) and isolated the region that contains the main IMP1 peak, which overlaps with m6A sites. Using a well-established system, we have inserted this region in a vector downstream of *Renilla* luciferase ORF. The vector also independently transcribes a firefly luciferase reporter which allows normalization of luciferase expression. We compared activity (i.e. protein expression) in the following contexts: i) IMP1 overexpression in wild type cells; ii) IMP1 overexpression in methyltransferase (METTL3) knock down cells; iii) IMP1 knock down, iv) METTL3 knock down; v) Both IMP1 and METTL3 knock down

While there is some experimental variability, as one can expect when doing co-transfection of both plasmids and siRNA, we show that IMP1 silencing leads to a decrease in activity, but the decrease is greater when both IMP1 and METTL3 are silenced. Importantly, overexpression of IMP1 leads to a several-fold increase in activity, but this increase is dependent on the expression of METTL3. These results i) show the m6A dependency of IMP1 regulation and, importantly, ii) are consistent with our model of m6A-mediated regulation of IMP1, where methylation provides an advantage (a few-fold increase in the bound protein) to an existing specific IMP1-RNA binding event. Next, we confirmed this with an equivalent assay for a second gene that we had also previously shown is regulated by both IMP1 and m6A in neurons (DCX), which strengthened our conclusions. These data are now discussed in the paper (**Fig. 3h-j**) where we write ‘**The role of m6A in IMP1-mediated gene regulation was further explored using a luciferase-based reporter assay in an established primary cell line which is more amenable to genetic manipulation. A portion of MAP2 3'UTR containing an IMP1 peak that overlapped with m6A sites was cloned downstream of *Renilla* luciferase ORF (**Fig. 3h, i**). We similarly cloned an IMP1-m6A peak in the DCX 3'UTR, to validate our observations. The luciferase activity was significantly reduced when vectors were co-transfected with an siRNA against IMP1, the m6A methyltransferase METTL3 or both. Notably, a concordant trend can be observed for both UTR RNAs, where the effect of IMP1 silencing is larger than that of the silencing of the methyltransferase. Cotransfection with both siRNAs showed a more pronounced effect than with each siRNA individually. This is consistent with our model where an existing IMP1 binding to the specific targets can be enhanced by m6A methylation, as we proposed based on the structure and biophysical characterisation of the IMP1 KH4-m6A interaction²³. Importantly, overexpression of IMP1 leads to an increase of expression which is abrogated when m6A is depleted (**Fig.3j**). The regulatory effect of m6A on IMP1 binding was validated in neurons by assessing the result of METTL3 KD on the regulated microtubules genes (**Fig.3k**).’.**

** -page 6, paragraph 4, "This suggests a mechanism whereby a lower protein concentration and a site-specific increase in affinity mediated by m6A methylation together increase IMP1 selection of neuronal (microtubule) targets (Fig.3j)": No data are

presented to support this specific mechanism. This model should be experimentally tested (see above).

This working mechanism is tested via a number of independent assays: 1) We have a structural and biophysical description of IMP1-m6A recognition and its comparison with the IMP1 recognition of non-methylated targets³. This study that indicates IMP1 can bind both unmodified and methylated RNAs and m6A methylation would enhance the binding to targets depending on the set of condition 2) We show that in a reporter gene system the regulation mediated by IMP1 interaction with target sites on microtubule mRNA are dependent on m6A methylation. We also observe a trend where the IMP+m6A KD is larger than the one of the individual KDs. This is consistent with a mechanism where methylation increases the affinity of an existing interaction. Noting that IMP1 targets with a binding advantage during neuronal differentiation, we also confirm that selected microtubule targets with IMP1-m6A peaks are regulated by both IMP1 and m6A in human neurons (**Fig. 3 h-k**) 3) We show that, at the transcriptome-wide level, IMP1 regulation of microtubule proteins is linked to the presence of m6A sites (**Fig. 3 e-g**).

-page 6, paragraph 4, "This is consistent with our recent structural and biophysical data showing that IMP1 KH4 directly recognizes m6A methylated RNA by IMP1 KH4 providing a few-fold increase in affinity and an advantage (Nicastro et al., unpublished data)": Is there a plan to include these data in the manuscript?

Please see the initial comment on the now published structural study on the principles of IMP1-m6A recognition. The concept we derived from the structural data is now discussed in the results section.

##Figures. (**: essential revisions)

-Fig. 1:

1) Fig. 1c: Please use a different color scheme with stronger contrast (ideally six distinct colors). With the current color scheme, the less abundant categories such as "intergenic" and "intron" are difficult to see. (They could be absent, but it is difficult to tell as shown.) The authors can find sample color schemes at colorbrewer2.org.

This has been done.

**2) Fig. 1d: Need to explain the two red lines in the legends and add an arrow or other indicators to guide the readers to "the peak adjacent to the stop codon" mentioned in the text.

This has now been explained in the legend and figure has been updated to make it clearer.

**3) Fig. 1e: The legends are missing.

This has now been corrected.

**4) Fig. 1f: Please explain/state the following in the legends: The numbers for the Y-

axis, the locations for the X-axis (last exon with 3' UTR?), and the categories (upregulated or stable) for each gene.

This has now been stated and the figure has been updated to make it clearer.

**5) Fig. 1h: The horizontal line below the asterisk is off. Please correct.

-Fig. 3:

This has now been corrected.

**1) Fig. 3h & 3i: Legends were swapped. Please correct.

This has now been corrected.

##Extended Data Figures. (**: essential revisions)

-Extended Data Fig. 2:

1) Appreciate the inclusion of the results from iCLIP optimization experiments.

**2) Fig. 2c: It looks like NPC#9 had the fewest unique counts but NPC#8 was excluded? Please explain or correct.

The reviewer is correct, NPC9 was excluded, not NPC8. This was a typographical error and has now been corrected.

-Extended Data Fig. 3:

**1) The legends does not match the figures. Please correct.

This has now been corrected.

2) All panels are a bit too small to read clearly.

Panels have been re-structured where possible.

-Extended Data Fig. 4:

1) Fig. 4f: Is the circle labeled "Nervous" the meta-analysis dataset? Please clarify in the legend. Maybe "Literature" or "Meta-analysis" would be better labels?

We agree with the reviewer. We have now re-label this figure as 'Neuron' (our dataset) and 'Meta-analysis', to clarify.

Reviewer #3 (Remarks to the Author):

In this paper, the authors aim to establish that, during the differentiation of human neurons, the RNA-binding protein IMP1 is a central regulator in the expression of mRNAs encoding major players in the microtubular network. They also aim to demonstrate the role of m6A methylation in the choice and selection of IMP1 target sites. This study is based essentially on the computational analysis of omics data obtained

from pluripotent stem cell cultures during neurogenesis: IMP1 CLIP, transcriptome, m6A RNA methylome, proteome.

The authors recently established that the reader IMP1 recognizes m6A marks independently of the sequence context (NAR 2023). This study brings functional meaning to this role of m6A reader in the context of neuronal differentiation. Overall, the hypothesis is appealing, the scientific approach is appropriate and the functional implications of this study could be significant for the field. The data obtained and their analysis seem to be of good quality (even if I am not an expert in bioinformatics). Nevertheless, some observations are mainly correlative and would benefit from more in-depth study, most particularly as the authors tackle an innovative concept. In my opinion, two important points are missing or need to be significantly strengthened:

We thank the reviewer for the comments and we address them below.

(1) The type of regulation exerted by IMP1 on target mRNAs, which remains speculative and would have benefited from a translactome analysis (by ribosome profiling or ribo-seq) and IMP1 CLIP from polysome fractions.

In the paper we show that IMP1 directly regulates the abundance of a large set of neuronal genes at the protein level. As commented by the reviewer, our results raise the interesting point of how IMP1 regulates gene expression at the protein level - it would be ideal to explore further this exciting finding. However, providing a mechanistic understanding of what is IMP1 regulation we feel would require a multi-layered set of new experiments, some of which are technically very challenging in neurons, and beyond the scope of this study, which focuses on the mechanism and effect of IMP1 target selection during neuronal development.

(2) As things stand, the data do not allow us to establish a direct link between m6A and IMP1. For this, it would be necessary to demonstrate (functionally) that the absence of the m6A mark leads to the loss of IMP1 binding at the sites concerned and the subsequent deregulation of corresponding protein. In this respect, does IMP1 play a direct/functional role on RNA or is it simply a competitor of m6A reader(s)?

We have now demonstrated this direct link using model genes and a reporter system. Please see answer to reviewer 2 above for the details of the experiments.

Other comments:

Fig 1b. It gives the wrong idea that IMP1 is only present in cell body in neurons. It would be nice to have a Zoom and show separate colors (in white for DAPI to make it more visible). Same for Sup Fig1B: DAPI staining is unclear.

We agree with the reviewer. Zoom images have now been added to Additional Material Fig.1

The authors establish that IMP1 iCLIP peaks are independent from mRNA concentrations. What about cell body vs neurite since IMP1 is involved in mRNA localization? Is there any data available on this?

Although we agree this would be an interesting extension of the paper, we do not have such analysis as such data have not yet been generated from human neurons, due to the technical challenges.

Fig 1d. Do the authors get a preferential distribution of IMP1 on CA-enriched motifs as previously published?

Yes, we do see one such sequence with a 30nt window. Please see the answer to the reviewer 2 above and related data.

Fig 1f. IMP1 distribution does not seem to be enriched in the 3'-UTR for these selected genes. Does it vary depending on mRNA? Is the distribution different in some specific groups (e.g. tubulin Network)?

We apologize for the lack of clarity. The box at the bottom of Figure 1f represents the 3'UTR, and IMP1 binds these targets in the 3'UTR as expected by the global distribution of binding sites. This is now labelled in the Figure and explained in the figure legend.

In addition, we have now conducted an analysis of IMP1 binding across various transcript regions within distinct Gene Ontology (GO) categories. Our findings indicate that, despite some variability in certain categories, the primary region of binding aligns with expectations, predominantly occurring in the 3'UTR. Please see the analysis below. Given this, we feel this analysis does not add substantially to the paper and we have, for now, left it out.

It seems to be the case since the authors say (p3) “this asymmetric distribution is exaggerated when considering the peaks of mRNAs encoding proteins involved in axonal development and microtubule organization,

Yes, this is indeed the case. The data are now shown in **Extended Data Fig.3b**, with the ratios and relevant GO terms reported in the legend. Please also see the answer to the relevant query of reviewer 2.

which our Gene Ontology analysis indicates are highly enriched in neuronal IMP1 targets.” Where is this analysis?

We apologize for the omission. The analysis is now shown in **Extended Data Fig.3c**

Fig 1h. There is a lot of variability of IMP1 protein expression in neurites. This is not the case in NPCs. One of the measures could well be an outlier (in particular the lowest one). The authors must show the WB in the extended data section.

We thank the reviewer for their comment, and we agree it is important to show representative WBs, particularly as the values for MNs WB were normalized against NPCs in the quantification in panel 1h. We have now added WB images for neurons and NPCs as **Extended data Fig. 3d**. Some variability in the values for both cell types is to be expected as we use independent lines (each derived from a different human individual) as biological replicates.

It is unclear to me why the authors focused on “microtubule” GO term rather than “myristylation” based on Fig 2c. Perhaps I missed the rationale behind this choice, even more considering that myristylation participates in cellular distribution of proteins in neurons. Could the authors comment on that?

We thank the reviewer for this insightful comment. Myristylation is quite a specific category with only four genes, three of which are regulated by IMP1. Terms that contain a very limited number of genes can be selected because an increase in one or two members of the gene set can lead to a disproportionate effect on the scores in a specific analysis. In the manuscript, we focused on microtubules because we wanted to explore the role of the protein in regulating a broad regulatory network and because this term is consistently present in neuron-specific targets, the targets with neuron-specific, expression-independent increase in binding and the regulated targets.

Is there a significant impact of IMP1 KD on neuron function/phenotype? It looks like neuron morphology is not altered at all (nor bIII tubulin staining) (ED Fig 3c).

We thank the reviewer for providing this comment. We have now added data and analysis that show IMP1 regulates neurite complexity and synaptic morphology, two crucial structural attributes of neurons that underlie function, in **Fig. 1i and Extended Data Fig.3e**

Minor remarks:

Fig 1: the legend for “1e” is missing

This has now been corrected.

Fig 3: “d” and “e” are mislabelled

This has now been corrected.

Define in the main text “IMP1-m6A peaks”

This has now been defined. ‘IMP1-m6A peaks - where IMP1 peaks overlap with an m6A site’

References for the rebuttal

1. Conway, A. E. *et al.* Enhanced CLIP Uncovers IMP Protein-RNA Targets in Human Pluripotent Stem Cells Important for Cell Adhesion and Survival. *Cell Rep.* **15**, 666–679 (2016).
2. Flamand, M. N., Tegowski, M. & Meyer, K. D. The Proteins of mRNA Modification: Writers, Readers, and Erasers. *Annu. Rev. Biochem.* **92**, 145–173 (2023).
3. Nicastro, G. *et al.* Direct m6A recognition by IMP1 underlays an alternative model of target selection for non-canonical methyl-readers. *Nucleic Acids Res.* **51**, 8774–8786 (2023).
4. Dagil, R. *et al.* IMP1 KH1 and KH2 domains create a structural platform with unique RNA recognition and re-modelling properties. *Nucleic Acids Res.* **47**, 4334–4348 (2019).
5. Patel, V. L. *et al.* Spatial arrangement of an RNA zipcode identifies mRNAs under post-transcriptional control. *Genes Dev.* **26**, 43–53 (2012).
6. Nicastro, G. *et al.* *Mechanism of [beta]-actin MRNA Recognition by ZBP1.* (Universität, 2017).
7. Mueller-Pillasch, F. *et al.* Expression of the highly conserved RNA binding protein KOC in embryogenesis. *Mech. Dev.* **88**, 95–99 (1999).
8. Hansen, T. V. O. *et al.* Dwarfism and impaired gut development in insulin-like growth factor II mRNA-binding protein 1-deficient mice. *Mol. Cell. Biol.* **24**, 4448–

4464 (2004).

REVIEWERS' COMMENTS

Reviewer #2 (Remarks to the Author):

The authors' efforts to address all the comments are appreciated, and the revised manuscript is substantially improved compared to the original one.

All my comments/concerns are satisfactorily addressed in the revised manuscript and I would like to recommend publication after a minor wording change listed below.

#Wording change:

It is understandable to use HeLa cells for the reporter assays but I find it confusing to describe HeLa cells as "an established primary cell line". Please directly state that these experiments were performed in HeLa cells in both the results (last paragraph) and the discussion (last paragraph) sections.

Reviewer #3 (Remarks to the Author):

I would like to thank the authors for this interesting work. All my concerns have been addressed.

Rebuttal to the reviewer's comments (Round 2).

We thank the reviewers' for their additional work. In order to address the only remaining comment, we now discuss the cells used in our reporter assays as HeLa.

Best regards,

Andres